# `M-Attack-V2`: Pushing the Frontier of Black-Box LVLM Attacks via Fine-Grained Detail Targeting

## Abstract

Black-box adversarial attacks on Large Vision–Language Models (LVLMs) present unique challenges due to the absence of gradient access and complex multimodal decision boundaries. While prior `M-Attack` demonstrated notable success with exceeding 90% attack success rate on GPT-4o/o1/4.5 by leveraging local crop-level matching between source and target data, we show this strategy introduces high-variance gradient estimates. Specifically, we empirically find that gradients computed over randomly sampled local crops are nearly orthogonal, violating the implicit assumption of coherent local alignment and leading to unstable optimization. To address this, we propose a theoretically grounded ***gradient denoising*** framework that redefines the adversarial objective as an expectation over local transformations. Our first component, *Multi-Crop Alignment (MCA)*, estimates the expected gradient by averaging gradients across diverse, independently sampled local transformations. This manner significantly reduces gradient variance, thus enhancing convergence stability. Recognizing an asymmetry in the roles of source and target transformations, we also introduce *Auxiliary Target Alignment (ATA)*. ATA regularizes the optimization by aligning the adversarial example not only with the primary target image but also with auxiliary samples drawn from a semantically correlated distribution. This constructs a smooth semantic trajectory in the embedding space, acting as a low-variance regularizer over the target distribution. Finally, we reinterpret prior momentum as replay through the lens of local matching as variance-minimizing estimators under the crop-transformed objective landscape. Momentum replay stabilizes and amplifies transferable perturbations by maintaining gradient directionality across local perturbation manifolds. Together, MCA, ATA, momentum replay, and a delicately selected ensemble set constitute `M-Attack-V2`, a principled framework for robust black-box LVLM attack. Empirical results show that our framework improves the attack success rate on GPT-4o (🌀) from **95%**→**99%**, on Claude-3.7 (✳) from **37%**→**67%**, and on Gemini-2.5-Pro (🔷) from **83%**→**97%**, significantly surpassing all existing black-box LVLM attacking methods.

## 1 Introduction

Large Vision-Language Models (LVLMs) have become foundational to modern AI systems, enabling multimodal tasks like image captioning [14, 34, 7, 37], VQA [27, 32], and visual reasoning [30]. However, their visual modules remain vulnerable to adversarial attacks, subtle perturbations that mislead models while remaining imperceptible to humans. Prior efforts, including AttackVLM [41], CWA [6], SSA-CWA [8], AdvDiffVLM [13], and most effectively, `M-Attack` [22], which have exploited this weakness through local-level matching and surrogate model ensembles, surpassing 90% success rates on models like GPT-4o.

Submitted to 39th Conference on Neural Information Processing Systems (NeurIPS 2025). Do not distribute.

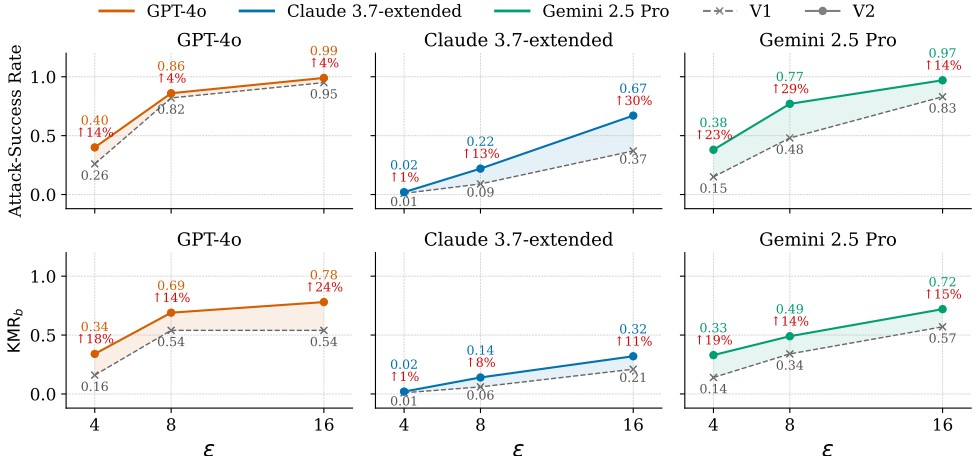

Figure 1: Improvment of `M-Attack-V2` over `M-Attack` on up-to-date commercial black-box models.

Despite its effectiveness, our analysis reveals that `M-Attack`'s gradient signals are highly unstable: Even overlapping large pixel regions, two consecutive local crops share *nearly orthogonal gradients*. In other words, high similarity in pixel and embedding space does not translate to high similarity in gradient space. The reason is that ViTs' gradient pattern is sensitive to translation. A tiny shift changes pixels contained in each token, altering self-attention. Moreover, patch-wise, spike-like gradient amplifies the mismatch within just a few pixels. We counter this effect by aggregating gradients from multiple crops within the same iteration, a strategy we call *Multi-Crop Alignment* (MCA). From a theoretical angle, MCA aggregates gradients across multiple views in a single iteration, smoothing local inconsistencies and improving cross-crop gradient stability.

We further observe that the source and target transformations in `M-Attack` operate in different semantic spaces: one emphasizing extraction, the other generalization. Aggressive target augmentation introduces harmful variance. Our *Auxiliary Target Alignment* (ATA) mitigates this by identifying semantically similar auxiliary images to create a low-variance embedding subspace, then applying only mild shifts to enhance transferability without destabilizing the optimization.

Classic momentum is reinterpreted under this framework as *Patch Momentum* (PM), a replay mechanism that recycles past gradients across random crops to stabilize optimization. In parallel, we also re-examine and enrich `M-Attack`'s model selection criterion and choose a delicately selected ensemble set with diverse patch sizes to mitigate the difficulty in cross-patch transfer, of which we find that the attention concentrates more on the main object. We term it *Patch Ensemble$^+$* (PE$^+$).

Together, these components, MCA, ATA, PM, and PE+, form the basis of `M-Attack-V2`, a robust gradient denoising framework that significantly outperforms existing black-box attack methods. Our method raises attack success rates from 95%→99% on GPT-4o, 37%→67% on Claude-3.7, and 83%→97% on Gemini-2.5-Pro, achieving state-of-the-art performance across the board. This study not only offers a practical, modular attack strategy but also sheds light on the gradient behavior of ViT-based LVLMs under local perturbations. We hope these insights will drive further research into transferable adversarial optimization under realistic black-box constraints.

## 2 Method

### 2.1 Limitations of Local Crop Matching in `M-Attack`

**Recall local matching framework in `M-Attack`.** To iteratively extract meaningful semantic details from the $\mathbf{X}_{\text{tar}}$ to $\mathbf{X}_{\text{sou}}$ (or called $\mathbf{X}_{\text{adv}}$), `M-Attack` proposes the local-level matching framework. Each region $\hat{\mathbf{x}}_i$ ($i \in \{1, 2, \ldots, n\}$) is generated independently at a different training iteration $i$:

$$\begin{aligned}
\{\hat{\mathbf{x}}_1^s, \ldots, \hat{\mathbf{x}}_n^s\} &= \mathcal{T}_s(\mathbf{X}_{\text{sou}}) \\
\{\hat{\mathbf{x}}_1^t, \ldots, \hat{\mathbf{x}}_n^t\}/\{\hat{\mathbf{x}}_g^t\} &= \mathcal{T}_t(\mathbf{X}_{\text{tar}}),
\end{aligned} \tag{1}$$

69 where $\mathcal{T}_s, \mathcal{T}_t$ are the set of random local mappings and subsequent preprocessing (i.e., crops and
70 resize) applied to the source and target images, respectively. $\hat{\mathbf{x}}_g^t$ is the globally transformed target
71 image across iterations. Without loss of generality, each pair $\hat{\mathbf{x}}_i^s$ and $\hat{\mathbf{x}}_i^t$ is matched in iteration $i$.

72 `M-Attack` introduces a local-matching strategy for attacking LVLMs in the black-box setting
73 by aligning spatial crops between source and target images. While effective, this approach
74 suffers from inherent instability in gradient-based optimization. Formally, consider a loss
75 function $\mathcal{L}(f(\mathcal{T}_s(\mathbf{X}_{\text{sou}})), f(\mathcal{T}_t(\mathbf{X}_{\text{tar}})))$, where $\mathcal{T}$ denotes a random local transformation (e.g., a
76 crop), $f$ is the white-box model like CLIP [33], and $\mathbf{X}_{\text{sou}}$ is the adversarial input. Because
77 $\nabla_{\mathbf{X}_{\text{sou}}}\mathcal{L}(f(\mathcal{T}_s(\mathbf{X}_{\text{sou}})), f(\mathcal{T}_t(\mathbf{X}_{\text{tar}})))$ varies significantly across $\mathcal{T}_s$ (same to $\mathcal{T}_t$), the stochastic gra-
78 dients become nearly orthogonal, i.e., $\langle \nabla_{\mathbf{X}_{\text{sou}}}\mathcal{L}_{\mathcal{T}_s^i}, \nabla_{\mathbf{X}_{\text{sou}}}\mathcal{L}_{\mathcal{T}_s^j}\rangle \approx 0$, leading to high variance and poor
79 convergence during optimization.

80 **Extremely low gradient overlap.** In `M-Attack` two random crops $\hat{\mathbf{x}}_i^s \subset \mathcal{T}_s(\mathbf{X}_{\text{sou}})$ and $\hat{\mathbf{x}}_i^t \subset \mathcal{T}_t(\mathbf{X}_{\text{tar}})$
81 are matched at every iteration. One would expect the gradients inside the shared region of two
82 successive source crops $(\hat{\mathbf{x}}_i^s, \hat{\mathbf{x}}_{i+1}^s)$ to correlate, because the underlying pixels partly coincide.
83 Supursingly, Fig. 2b shows the opposite: their cosine similarity is **almost zero**. We then keep one
84 crop fixed and vary the other across scales and IoUs (Fig. 2a). Our finding reveals an exponential
85 decay that plateaus below $0.1$ once the overlap is smaller than $0.80$ IoU.

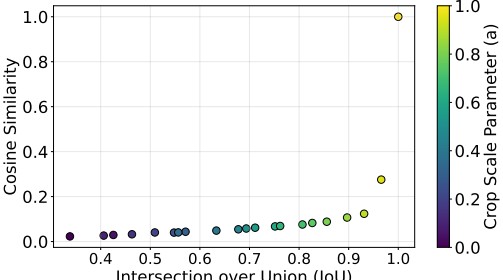 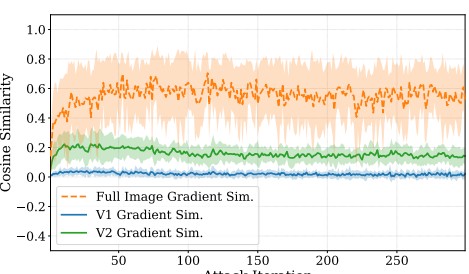

(a) Similarity over IoU. The results are averaged from 20 runs with different crop parameter $a$ for $[a, 1.0]$.

(b) Comparison of gradient similarity from full image update and local matching over each iteration

Figure 2: Similarities of gradients from different crops. a) similarity over IoU for different crops by fixing in one iteration; b) similarity between two consecutive gradients across iterations. Results are averaged from 200 runs.

86 **Source.** We find two main reasons behind this high variance: ViT's inherent sensitivity to translation
87 and asymmetry within the local matching framework. We discuss them below.

88 *Patch-wise, spike-like gradient sensitive to translation.* Because ViTs tokenize images on a fixed,
89 non-overlapping grid, even sub-pixel changes each patch's token mix. These token changes ripple
90 through self-attention, altering weights and redirecting gradients for *all* tokens, so the resulting
91 pixel-level gradient pattern diverges sharply. Worse, gradient magnitudes are uneven. Therefore, even
92 similar patterns but missing a few pixels might break gradient similarity (Fig. 3b).
93 *Asymmetric Transform Branches.* In `M-Attack`, both the *source* and *target* images are cropped, yet
94 playing distinct roles. Cropping the source acts directly in *pixel space*: it rearranges patch embeddings
95 and attention weights in the forward pass, ending up with guidance of different views. By contrast,
96 cropping the target sorely translate the target representation, thereby shifting the reference embedding
97 in *feature space*. One sculpts the perturbation, while another moves the goalpost, formulating
98 asymmetric matching. `M-Attack` overlooked this and implementations target translation alternate
99 between a *radical* crop and an identity map, struggles between explore-exploitation trade-off and
100 potentially risk in high variance of target embedding.

101 **Asymmetric Matching over Expectation.** To mitigate the issues above, we begin by concisely
102 reformulating the original objective function as an expectation over local transformations within an
103 asymmetric matching framework:

$$\min_{\|\mathbf{X}_{\text{sou}}\|_p \leq \epsilon} \mathbb{E}_{\mathcal{T}\sim\mathcal{D}, y\sim\mathcal{Y}}\left[\mathcal{L}(f(\mathcal{T}(\mathbf{X}_{\text{sou}})), \mathbf{y})\right], \tag{2}$$

104 d where $\mathcal{D}$ represents the distribution of local transformations, and $\mathcal{Y}$ denotes the distribution over
105 target semantics. $\|\cdot\|_p$ is $\ell_p$ constraint for imperceptibility. Conceptually, this formulation corresponds
106 to embedding specific semantic content $y$ into a locally transformed area $\mathcal{T}(\mathbf{X}_{\text{sou}})$, thus highlighting

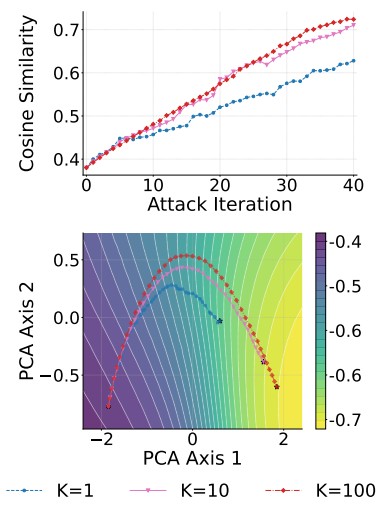

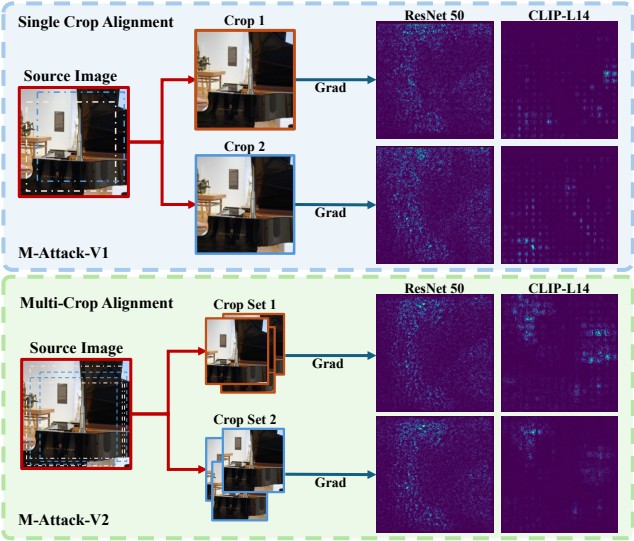

(a) Comparison of optimization trajectories with different $K$, $K = 1$ refers to single crop alignment.

(b) Gradient pattern between different crop strategies in M-Attack and M-Attack-V2.

Figure 3: Comparison of: a) different trajectories against different $K$; b) gradient pattern of single crop alignment against multi-crop alignment (MCA). The gradient pattern of ResNet 50 remains consistent when large pixels are overlapped, while the gradient pattern of ViTs changes dramatically. MCA helps to smooth out this impact.

the intrinsic asymmetry compared to M-Attack's original formulation. Within this framework, our proposed enhancements, i.e., *Multi-Crop Alignment* (MCA) and *Auxiliary Target Alignment* (ATA), can be interpreted as strategies to improve the accuracy of the expectation estimation and the sampling quality of the semantic distribution $\mathcal{Y}$.

## 2.2 *Gradient Denoising* via **Multi-Crop Alignment (MCA)**

To obtain a low-variance estimate of the expected loss gradient $\mathbb{E}_{\mathcal{T} \sim \mathcal{D}, y \sim \mathcal{Y}} [\nabla_{\mathbf{x}_{\text{sou}}} \mathcal{L}(f(\mathcal{T}(\mathbf{X}_{\text{sou}})), \mathbf{y})]$, we draw $K$ independent crops $\{\mathcal{T}\}_{k=1}^{K}$ and average their individual gradients:

$$\nabla_{\mathbf{x}_{\text{sou}}} \hat{\mathcal{L}}(\mathbf{X}_{\text{sou}}) = \frac{1}{K} \sum_{k=1}^{K} \nabla_{\mathbf{x}_{\text{sou}}} \mathcal{L}(f(\mathcal{T}_k(\mathbf{X}_{\text{sou}})), \mathbf{y}). \quad (3)$$

This *Multi-Crop Alignment (MCA)* acts as an unbiased Monte-Carlo estimator, thus naturally reducing the variance with $K > 1$.

**Theorem 1.** *Let $g_k = \nabla_{\mathbf{x}_{\text{sou}}} \mathcal{L}(f(\mathcal{T}_k(\mathbf{X}_{\text{sou}})), y)$ denote the gradient from $\mathcal{T}_k$, $\mu = \mathbb{E}[g_k]$, $\sigma^2 = \mathbb{E}[\|g_k - \mu\|_2^2]$ denote the mean and variance, and $p_{k\ell}$ denote the pair-wise correlation $p_{k\ell} = \frac{\langle g_k - \mu, g_\ell - \mu \rangle}{\|g_k - \mu\|^2 \|g_\ell - \mu\|^2}$. The gradient variance from $K$ averaged crops is bounded by*

$$\text{Var}\left(\frac{1}{K} \sum_{k=1}^{K} g_k\right) \leq \frac{\sigma^2}{K} + \frac{K-1}{K} \overline{p} \sigma^2, \quad (4)$$

*where $\overline{p} = \mathbb{E}[p_{kl}]$, $k \neq \ell$ is the expectation of pair-wise correlation*

All crops share the same underlying image, so $\overline{p} \neq 0$. The ideal $\sigma^2/K$ decay is therefore tempered by the correlation term $\overline{p}\sigma^2$. Empirically, averaging a modest number ($K = 10$) of almost-orthogonal gradients still yields benefit, since the uncorrelated component of the variance shrinks as $1/K$. Simultaneously, the optimizer leverages multiple diverse transformations per update, with minimal interference among almost orthogonal gradients. Fig. 3a illustrates an accelerated convergence with $K = 10$, with margin improvement provided by $K = 100$.

This averaging also alleviates the known translation sensitivity of ViTs. As shown in Fig. 3b, using two crop sets yields noticeably higher gradient consistency than the single-crop alignment in `M-Attack`. In MCA, high-activity regions remain stable (upper left and center right), while the single-crop case shifts focus from center right to lower left. As a result, gradient similarity across iterations increases from near zero in `M-Attack` to around 0.2 (Fig. 2b).

### 2.3 *Improved Sampling Quality* via Auxiliary Target Alignment (ATA)

Selecting a representative target embedding $y \in \mathcal{Y}$ is challenging because the underlying distribution $\mathcal{Y}$ is not observable. `M-Attack` mitigates this by seeding at the unaltered target embedding $f(\mathbf{X}_{\text{tar}})$ and exploring its vicinity with transformed views $f(\mathcal{T}_t(\mathbf{X}_{\text{tar}}))$ thereby sketching a locally semantic manifold that serves as a proxy for $\mathcal{Y}$. However, the exploration–exploitation trade-off remains problematic. *Radical* transformations leap too far, dragging $y$ outside the genuine target region; *conservative* transformations, while semantically faithful, barely shift the embedding, leaving the optimization starved of informative signal.

To stabilize this process, we introduce $P$ auxiliary images $\{\mathbf{X}_{\text{aux}}^{(p)}\}_{p=1}^P$ that act as additional anchors, collectively forming a richer sub-manifold of aligned embeddings. During each update, we apply a *mild* random transformation $\tilde{\mathcal{T}} \sim \tilde{\mathcal{D}}$ to every anchor, nudging the ensemble in a coherent yet restrained manner and thus providing low-variance, information-rich gradients for optimization. Let $y_0 = f(\hat{\mathcal{T}}_0(\mathbf{X}_{\text{tar}})), \tilde{y}_p = f(\tilde{\mathcal{T}}_p(\mathbf{X}_{\text{aux}}^{(p)}))$ denote sampled semantics in one iteration. The objective $\hat{\mathcal{L}}$ in Equ. (3) becomes

$$\hat{\mathcal{L}} = \frac{1}{K} \sum_{k=1}^{n} \left[ \mathcal{L}(f(\mathcal{T}_k(\mathbf{X}_{\text{sou}})), y_0) + \frac{\lambda}{P} \sum_{p=1}^{P} \mathcal{L}(f(\mathcal{T}_k(x)), \tilde{y}_p) \right] \tag{5}$$

where $\lambda \in [0, 1]$ interpolates between the original target and its auxiliary neighbors. $\lambda = 0$ reduce to `M-Attack` local-local matching with single target. ATA trade-off exploration (auxiliary diversity) and exploitation (main-target fidelity), providing low-variance, semantics-preserving updates. The auxiliary set can be built variously, i.e., through image-image retrieval or diffusion methods.

**Cost.** Each iteration back-propagates through the $K$ source crops and only forward-propagates the $P$ auxiliary targets. Since a backward pass is roughly twice as expensive as a forward pass, the per-iteration complexity is $\mathcal{O}\big(K(3+P)\big)$, roughly twice the overhead when $P = 3$.

### 2.4 Patch Momentum with Built-in Replay Effect

Momentum, introduced in MI-FGSM [11], is widely adopted to improve the transferability. Define the momentum buffer as: $m_r = \beta_1 m_{r-1} + (1 - \beta_1)\nabla_{\hat{\mathbf{x}}^s}\hat{\mathcal{L}}_r(\hat{\mathbf{x}}^s)$, where $\beta_1 \in [0, 1)$ is the first-order momentum coefficient and $\nabla_{\hat{\mathbf{x}}^s}\hat{\mathcal{L}}_r(\hat{\mathbf{x}}^s)$ is our MCA-ATA-estimated gradient $g_r$ at iteration $r$.

Under the local-matching view, this mechanism can be reinterpreted as formulating a streaming MCA to enforce temporal consistency across gradient directions in the space of random crops. Unrolling the EMA for pixel $k$ exposes an alternative interpretation:

$$m_i(k) = (1 - \beta) \sum_{j=0}^{i} \beta^j \, \mathbf{1}\{k \in M_{i-j}\} g_{i-j}(k), \tag{6}$$

where $M_i$ denotes the pixel indices included in iteration $i$, $m_i(k)$ and $g_i(k)$ respectively denotes momentum and gradient for pixel $k$. Each crop that involves pixel $k$ is therefore replayed in future iterations with geometrically decaying weight, allowing rarely sampled regions (such as corners) to persist long enough to combat the gradient starvation. Spike-shaped gradients are further moderated by the Adam-style [18] second moment, $v_r = \beta_2 v_{r-1} + (1 - \beta_2)g_r^2$,, whose scaling effect is essential in our empirical study. The momentum does not directly improve gradient similarity but continuously re-injects historical crops across patches, effectively maintaining gradient directionality across local perturbation manifolds. We therefore term it *Patch Momentum* to distinguish.

The whole procedure, combining MCA, ATA, and PM, is detailed in Alg. 1. We use a different color to differentiate between `M-Attack-V2` and `M-Attack`. We use PGD [29] with ADAM [18] for line 13. The appendix presents analogous results for FGSM and I-FGSM variants.

**Algorithm 1** `M-Attack-V2`

---

**Require:** clean image $\mathbf{X}_{\text{clean}}$; primary target $\mathbf{X}_{\text{tar}}$; auxiliary set $\mathcal{A} = \left\{\mathbf{X}_{\text{aux}}^{(p)}\right\}_{p=1}^{P}$; patch ensemble$^{+}$
$\quad \Phi^{+} = \{\phi_j\}_{j=1}^{m}$; iterations $n$, step size $\alpha$, perturbation budget $\epsilon$; number of crops $K$, auxiliary weight
$\quad \lambda \ (0 \le \lambda \le 1)$;
1: $\mathbf{X}_{\text{adv}} \leftarrow \mathbf{X}_{\text{clean}}$,
2: **for** $i = 1$ **to** $n$ **do**
3: $\quad$ Draw $K$ transforms $\{\mathcal{T}_k\}_{k=1}^{K} \sim \mathcal{D}$
4: $\quad g \leftarrow \mathbf{0}$ $\hspace{5.5cm}$ ▷ accumulate over crops
5: $\quad$ **for** $k = 1$ **to** $K$ **do** $\hspace{5cm}$ ▷ — crop loop —
6: $\quad\quad$ Draw $\{\tilde{\mathcal{T}}_p\}_{p=0}^{P} \sim \tilde{D}$
7: $\quad\quad$ **for** $j = 1$ **to** $m$ **do**
8: $\quad\quad\quad y_0 = f(\tilde{\mathcal{T}}_p(\mathbf{X}_{\text{tar}})), \ y_p = f(\tilde{\mathcal{T}}_p(\mathbf{X}_{\text{aux}}^{(p)})), p = 1, \dots, P$ $\quad$ ▷ Transform target and auxiliary data
9: $\quad\quad\quad$ Compute $\hat{\mathcal{L}}_k = (f_{\phi_j}(\mathcal{T}_k(\mathbf{X}_{\text{sou}})), y_0) + \frac{\lambda}{P}\sum_{p=1}^{P}\mathcal{L}(f_{\phi_j}(\mathcal{T}_k(x)), \tilde{y}_p)$
10: $\quad\quad\quad g \leftarrow g + \frac{1}{Km}\nabla_{\mathbf{x}_{\text{sou}}}\hat{\mathcal{L}}_k$
11: $\quad\quad$ **end for**
12: $\quad$ **end for**
13: $\quad$ Updated $\mathbf{X}_{\text{adv}}$ based on $g$ with Patch Momentum
14: **end for**
15: **return** $\mathbf{X}_{\text{adv}}$

---

## 3 Experiments

### 3.1 Experimental Setup

**Metrics.** We follow the evaluation protocol of `M-Attack`, reporting the *Attack Success Rate* (ASR) computed with *GPTScore* and the *Keywords Matching Rate* (KMR) at three thresholds $\{0.25, 0.5, 1.0\}$, denoted as $\text{KMR}_a$, $\text{KMR}_b$, and $\text{KMR}_c$ [22]. KMR leverages human-annotated semantic keywords and measures different levels of keywords matching, treating the matching rate greater than $x$ as a successful attack, denoting the final success rate $\text{KMR}_x$. The evaluation prompt and the keyword sets are identical to those in `M-Attack`.

**Surrogate candidates.** We adopt the exact surrogate selections used in their original papers for ensemble-based baselines [40, 8, 13, 22]. Our candidate pool includes CLIP series (CLIP-B/16, CLIP-B/32, CLIP$^{†}$-G/14[1] CLIP$^{†}$-B/32, CLIP$^{†}$-H/14, CLIP-L/14, CLIP$^{†}$-B/16, CLIP$^{†}$-BG/14), DinoV2 family [31] (Dino-Small, Dino-Base, Dino-Large), and the shared vision encoder of BLIP-2 family [20]. See the appendix for more details.

**Victim black-box models and dataset.** We evaluate four cutting-edge commercial multimodal LLMs: GPT-4o [1], o3 [30], Claude-3.7-Sonnet-extended [3], and Gemini-2.5-Pro-Preview [36]. Clean images are drawn from the *NIPS 2017 Adversarial Attacks and Defenses Competition* dataset [17]. Following SSA-CWA [9] and `M-Attack` [22], we randomly sample 100 images. Auxiliary sets are retrieved from the COCO training set [23] using CLIP-B/16 embedding similarity. Further results on a 1k image subset are in the appendix.

**Hyperparameters.** Unless otherwise noted, perturbations are bounded by $\ell_{\infty}$ with $\epsilon = 16$ and optimized for 300 steps. We set the step size to $\alpha = 0.75$ for Claude and $\alpha = 1.0$ for all other victims, mirroring `M-Attack`. Our `M-Attack-V2` attack utilizes, $\alpha = 1.275$, $\beta_1 = 0.9$, $\beta_2 = 0.999$ for momentum $K = 10$, $P = 2$, and $\lambda = 0.3$ for MCA and ATA. Ablation on $\alpha$ is in the appendix. The target transformation $\tilde{\mathcal{T}}$ includes random resized crop ($[0.9, 1.0]$), random horizontal flip ($p = 0.5$), and random rotation ($\pm 15°$).

### 3.2 Selection of surrogate model

Ensembling surrogate models is typical for enhancing black-box adversarial transferability. To further improve, advanced gradient aggregation methods [40, 13] are proposed, yet another practical and efficient way parallel to aggregation is to select models strategically.

We first profile the embedding transferability on different surrogate models, presented in Tab. 1. Results show that cross-model, especially cross-patchsize transfer, is difficult. Therefore, we retain models with diverse patch sizes that perform well in Tab 1. Trails of different combinations in

---

[1]$^{†}$ denotes trained on LAION [35] dataset

the appendix yield our *Patch Ensemble*$^+$(PE$^+$), comprising *CLIP$^\dagger$-G/14, CLIP-B/16, CLIP-B/32, and CLIP$^\dagger$-B/32*. Attention maps reveal a possible explanation: PE$^+$ models tend to concentrate attention on the main object, whereas others exhibit dispersed focus across unrelated regions. We hypothesize that focusing on the main object enhances transferability, as all models share the common objective of identifying core semantic content. In contrast, attention to scattered regions may capture model-specific biases that do not generalize well across architectures.

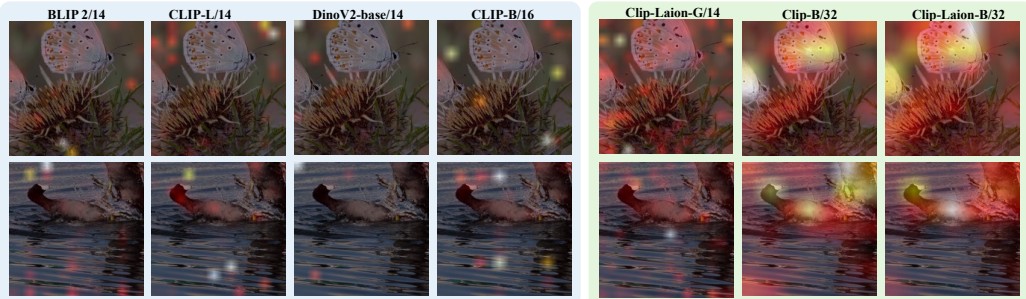

Figure 4: Comparison of two types of attention maps. Left: attention map that sparsely separates in different regions; right: attention map that focus to the main object.

| Surrogate | C–L/14 | C$^\dagger$–L/14 | D–S/14 | D–B/14 | D–L/14 | C–B/16 | C$^\dagger$–B/16 | C–B/32 | C$^\dagger$–B/32 | BLIP2 | Avg/14 | Avg/16 | Avg/32 | Avg/All |
|---|---|---|---|---|---|---|---|---|---|---|---|---|---|---|
| C–L/14 | N/A | 0.40 | 0.10 | 0.13 | 0.12 | 0.45 | 0.40 | 0.34 | 0.24 | 0.48 | 0.25 | 0.42 | 0.29 | 0.30 |
| C$^\dagger$–L/14 | 0.44 | N/A | 0.24 | 0.24 | 0.21 | 0.55 | 0.57 | 0.37 | 0.33 | 0.61 | 0.35 | 0.56 | 0.35 | 0.39 |
| D–S/14 | 0.25 | 0.39 | N/A | 0.45 | 0.38 | 0.41 | 0.45 | 0.32 | 0.25 | 0.46 | 0.39 | 0.43 | 0.28 | 0.37 |
| D–B/14 | 0.29 | 0.36 | 0.33 | N/A | 0.51 | 0.37 | 0.39 | 0.31 | 0.23 | 0.47 | 0.39 | 0.38 | 0.27 | 0.36 |
| D–L/14 | 0.26 | 0.31 | 0.12 | 0.32 | N/A | 0.31 | 0.34 | 0.30 | 0.21 | 0.42 | 0.29 | 0.33 | 0.26 | 0.29 |
| C–B/16 | 0.44 | 0.43 | 0.21 | 0.18 | 0.13 | N/A | 0.53 | 0.37 | 0.27 | 0.51 | 0.32 | 0.53 | 0.32 | 0.34 |
| C$^\dagger$–B/16 | 0.43 | 0.51 | 0.22 | 0.21 | 0.15 | 0.57 | N/A | 0.39 | 0.34 | 0.52 | 0.34 | 0.57 | 0.36 | 0.37 |
| C–B/32 | 0.37 | 0.43 | 0.21 | 0.11 | 0.09 | 0.55 | 0.53 | N/A | 0.49 | 0.46 | 0.28 | 0.54 | 0.49 | 0.36 |
| C$^\dagger$–B/32 | 0.31 | 0.49 | 0.27 | 0.18 | 0.12 | 0.53 | 0.61 | 0.58 | N/A | 0.50 | 0.31 | 0.57 | 0.58 | 0.40 |
| BLIP2 | 0.39 | 0.43 | 0.15 | 0.20 | 0.26 | 0.45 | 0.43 | 0.33 | 0.25 | N/A | 0.29 | 0.44 | 0.29 | 0.32 |

Table 1: Comparison of embedding transferability over 1k images. MCA/ATA excluded to show standalone performance. C/D = CLIP/DinoV2. Gray denotes selected models.

| Method | Model | GPT-4o | | | | Claude 3.7-extended | | | | Gemini 2.5-Pro | | | | Imperceptibility | |
|---|---|---|---|---|---|---|---|---|---|---|---|---|---|---|---|
| | | KMR$_a$ | KMR$_b$ | KMR$_c$ | ASR | KMR$_a$ | KMR$_b$ | KMR$_c$ | ASR | KMR$_a$ | KMR$_b$ | KMR$_c$ | ASR | $\ell_1\downarrow$ | $\ell_2\downarrow$ |
| AttackVLM [41] | B/16 | 0.09 | 0.04 | 0.00 | 0.02 | 0.04 | 0.02 | 0.00 | 0.00 | 0.08 | 0.04 | 0.00 | 0.00 | 0.034 | 0.040 |
| | B/32 | 0.07 | 0.03 | 0.00 | 0.03 | 0.06 | 0.04 | 0.00 | 0.01 | 0.09 | 0.05 | 0.00 | 0.02 | 0.036 | 0.041 |
| | Laion$^\dagger$ | 0.07 | 0.04 | 0.00 | 0.02 | 0.05 | 0.02 | 0.04 | 0.01 | 0.09 | 0.05 | 0.00 | 0.01 | 0.035 | 0.040 |
| AdvDiffVLM [13] | Ensemble | 0.02 | 0.00 | 0.00 | 0.02 | 0.01 | 0.00 | 0.00 | 0.01 | 0.03 | 0.01 | 0.00 | 0.00 | 0.064 | 0.095 |
| SSA-CWA [8] | Ensemble | 0.11 | 0.06 | 0.00 | 0.09 | 0.06 | 0.04 | 0.01 | 0.12 | 0.05 | 0.03 | 0.01 | 0.08 | 0.059 | 0.060 |
| AnyAttack [40] | Ensemble | 0.44 | 0.20 | 0.04 | 0.42 | 0.19 | 0.08 | 0.01 | 0.22 | 0.35 | 0.06 | 0.01 | 0.34 | 0.048 | 0.052 |
| M-Attack [22] | Ensemble | 0.82 | 0.54 | 0.13 | 0.95 | 0.31 | 0.21 | 0.04 | 0.37 | 0.81 | 0.57 | 0.15 | 0.83 | **0.030** | **0.036** |
| M-Attack-V2 (Ours) | Ensemble | **0.91** | **0.78** | **0.40** | **0.99** | **0.56** | **0.32** | **0.11** | **0.67** | **0.87** | **0.72** | **0.22** | **0.97** | 0.038 | 0.044 |

Table 2: Comparison of M-Attack-V2 with other black-box LVLM attack methods.

### 3.3 Extensive Evaluation Across LVLMs and Settings

**Transferability across LVLMs.** Tab. 2 illustrates the superiority of our M-Attack-V2 compared to the other black-box LVLM attack method. Our method leads others by a large margin, including M-Attack. On GPT-4o and Geimin 2.5-Pro, our M-Attack-V2 even achieves ASR close to 100%, with ASR on Claude 3.7-extended further improved by 30%, which is difficult for M-Attack to attack. Note that these improvements come with a slight increase in the perturbation norms for $l_1$ and $l_2$. Previous $l_1$ and $l_2$ norms are caused by insufficient optimization through near-orthogonal gradients; thus, the perturbation norm only increases in a sub-linear pattern. Our M-Attack-V2 mitigates this issue, exploring more sufficiently inside the $l_\infty$ ball. Thus, it slightly increases the perturbation magnitude. See the appendix for visualizations of these adversarial samples.

**Performance under budget constraints.** Tab. 3 compares performance under varying perturbation budgets ($\epsilon$). Our method consistently ranks among the top, achieving the best or second-best results across all settings. Notably, the margin is substantial when it leads, demonstrating its superior ability to explore within different $\ell_\infty$ balls.

Fig. 5 compares performance under different optimization budgets (total steps). Our method converges faster than M-Attack, reaching near-optimal results by 300 steps. In contrast, M-Attack continues to

improve with an additional 200 steps, indicating slower convergence. At 100 and 200 steps, `M-Attack`
shows a significant performance drop, while our method maintains more stable ASR and $KMR_b$. This
robustness stems from reduced variance, as `M-Attack` is more affected by random cropping on the
source and radical transformations on the target image, requiring more iterations to stabilize.

**Robustness Against Vision-Reasoning Models.** We further evaluate `M-Attack-V2` against GPT-o3,
a model enhanced with visual reasoning capabilities. As shown in Tab. 5, GPT-o3 exhibits slightly
better robustness than GPT-4o. However, the limited improvement suggests that its reasoning module
is not explicitly trained to detect adversarial manipulations. Thus, even after reasoning, GPT-o3
remains susceptible to `M-Attack-V2`. Its reasoning process is presented in the appendix.

| $\epsilon$ | Method | GPT-4o | | | | Claude 3.7-thinking | | | | Gemini 2.5-Pro | | | | Imperceptibility | |
|---|---|---|---|---|---|---|---|---|---|---|---|---|---|---|---|
| | | $KMR_a$ | $KMR_b$ | $KMR_c$ | ASR | $KMR_a$ | $KMR_b$ | $KMR_c$ | ASR | $KMR_a$ | $KMR_b$ | $KMR_c$ | ASR | $\ell_1\downarrow$ | $\ell_2\downarrow$ |
| 4 | AttackVLM [41] | 0.08 | 0.04 | 0.00 | 0.02 | 0.04 | 0.01 | 0.00 | 0.00 | 0.10 | 0.04 | 0.00 | 0.01 | 0.010 | 0.011 |
| | SSA-CWA [8] | 0.05 | 0.03 | 0.00 | 0.03 | 0.04 | 0.01 | 0.00 | 0.02 | 0.04 | 0.01 | 0.00 | 0.04 | 0.015 | 0.015 |
| | AnyAttack [40] | 0.07 | 0.02 | 0.00 | 0.05 | 0.05 | **0.05** | **0.02** | **0.06** | 0.05 | 0.02 | 0.00 | 0.10 | 0.014 | 0.015 |
| | M-Attack [22] | 0.30 | 0.16 | 0.03 | 0.26 | 0.06 | 0.01 | 0.00 | 0.01 | 0.24 | 0.14 | 0.02 | 0.15 | **0.009** | **0.010** |
| | M-Attack-V2 (Ours) | **0.59** | **0.34** | **0.10** | **0.58** | 0.06 | 0.02 | 0.00 | 0.02 | **0.48** | **0.33** | **0.07** | **0.38** | 0.012 | 0.013 |
| 8 | AttackVLM [41] | 0.08 | 0.02 | 0.00 | 0.01 | 0.04 | 0.02 | 0.00 | 0.01 | 0.07 | 0.01 | 0.00 | 0.01 | 0.022 | 0.022 |
| | SSA-CWA [8] | 0.06 | 0.02 | 0.00 | 0.04 | 0.04 | 0.02 | 0.00 | 0.04 | 0.02 | 0.00 | 0.00 | 0.05 | 0.030 | 0.030 |
| | AnyAttack [40] | 0.17 | 0.06 | 0.00 | 0.13 | 0.07 | 0.07 | 0.02 | 0.05 | 0.12 | 0.04 | 0.00 | 0.13 | 0.028 | 0.029 |
| | M-Attack [22] | 0.74 | 0.50 | 0.12 | 0.82 | 0.12 | 0.06 | 0.00 | 0.09 | 0.62 | 0.34 | 0.08 | 0.48 | **0.017** | **0.020** |
| | M-Attack-V2 (Ours) | **0.87** | **0.69** | **0.20** | **0.93** | **0.23** | **0.14** | **0.02** | **0.22** | **0.72** | **0.49** | **0.21** | **0.77** | 0.023 | 0.023 |
| 16 | AttackVLM [41] | 0.08 | 0.02 | 0.00 | 0.02 | 0.01 | 0.00 | 0.00 | 0.01 | 0.03 | 0.01 | 0.00 | 0.00 | 0.036 | 0.041 |
| | SSA-CWA [8] | 0.11 | 0.06 | 0.00 | 0.09 | 0.06 | 0.04 | 0.01 | 0.12 | 0.05 | 0.03 | 0.01 | 0.08 | 0.059 | 0.060 |
| | AnyAttack [40] | 0.44 | 0.20 | 0.04 | 0.42 | 0.19 | 0.08 | 0.01 | 0.22 | 0.35 | 0.06 | 0.01 | 0.34 | 0.048 | 0.052 |
| | M-Attack [22] | 0.82 | 0.54 | 0.13 | 0.95 | 0.31 | 0.21 | 0.04 | 0.37 | 0.81 | 0.57 | 0.15 | 0.83 | **0.030** | **0.036** |
| | M-Attack-V2 (Ours) | **0.91** | **0.78** | **0.40** | **0.99** | **0.56** | **0.32** | **0.11** | **0.67** | **0.87** | **0.72** | **0.22** | **0.97** | 0.038 | 0.044 |

Table 3: Ablation study on the impact of perturbation budget ($\epsilon$).

| Component | | | Gemini 2.5-Pro | | | | Claude 3.7-extended | | | |
|---|---|---|---|---|---|---|---|---|---|---|
| MCA | ATA | PM | $KMR_a$ | $KMR_b$ | $KMR_c$ | ASR | $KMR_a$ | $KMR_b$ | $KMR_c$ | ASR |
| | | | 0.87 | 0.72 | 0.22 | 0.97 | 0.56 | 0.32 | 0.11 | 0.67 |
| ✗ | | | 0.85 | 0.70 | 0.21 | 0.92 | 0.52 | 0.35 | 0.08 | 0.66 |
| | | | ↓0.02 | ↓0.02 | ↓0.01 | ↓0.05 | ↓0.04 | ↑0.03 | ↓0.03 | ↓0.01 |
| | ✗ | | 0.85 | 0.68 | 0.21 | 0.93 | 0.55 | 0.22 | 0.10 | 0.62 |
| | | | ↓0.02 | ↓0.04 | ↓0.01 | ↓0.04 | ↓0.01 | ↓0.10 | ↓0.01 | ↓0.05 |
| ✗ | ✗ | | 0.82 | 0.62 | 0.22 | 0.93 | 0.44 | 0.31 | 0.08 | 0.62 |
| | | | ↓0.05 | ↓0.10 | – | ↓0.04 | ↓0.12 | ↓0.01 | ↓0.03 | ↓0.05 |
| | | ✗* | 0.82 | 0.71 | 0.21 | 0.96 | 0.52 | 0.32 | 0.10 | 0.66 |
| | | | ↓0.05 | ↓0.01 | ↓0.01 | ↓0.01 | ↓0.04 | ↓0.00 | ↓0.01 | ↓0.01 |
| | | ✗ | 0.39 | 0.23 | 0.08 | 0.35 | 0.07 | 0.03 | 0.00 | 0.08 |
| | | | ↓0.48 | ↓0.49 | ↓0.14 | ↓0.62 | ↓0.49 | ↓0.29 | ↓0.11 | ↓0.59 |

Table 4: Effect of removing each component. Numbers below each value denote the change relative to the full model (first row). ✗ marks the component(s) disabled. * removes *only* the first-order term.

| Model | $KMR_a$ | $KMR_b$ | $KMR_c$ | ASR |
|---|---|---|---|---|
| GPT-o3 (o3-2025-04-16) | 0.91 | 0.71 | 0.23 | 0.98 |

Table 5: Results of `M-Attack-V2` on vision reasoning model

Figure 5: Comparison of different methods under different step budgets.

### 3.4 Ablation Study

**Contribution of each component.** Tab. 4 presents the performance changes when removing each
component. Results on GPT-4o are excluded due to non-significant differences. Both MCA and
ATA contribute approximately 5% improvement. Removing first-order momentum causes a slight
drop, while eliminating both first- and second-order momentum leads to a substantial decline. This
highlights the importance of second-order momentum in the PGD framework, as it helps normalize
the varying gradient scales in ViTs, potentially enhancing alignment.

**Hyperparameter.** Fig. 6 (left) shows that transferability initially improves with increasing $K$,
then declines. The optimal $K$ lies around $10 \sim 20$. Moderate noise helps escape local minima
and enhances transferability. However, as $K$ grows, training becomes more stable but loses this
regularizing effect. Fig. 6 (right) illustrates the impact of $\lambda$ on the transferability. Larger $\lambda$ provides

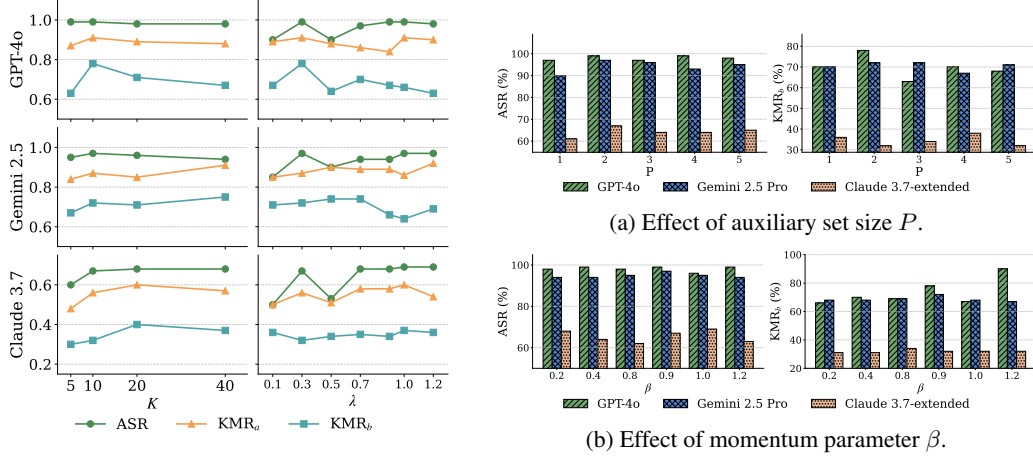

Figure 6: ASR and KMR$_a$/KMR$_b$ vs. different $K$ and $\lambda$.

Figure 7: Ablation study on auxiliary set size $P$ and momentum parameter $\beta$.

better diversity, drags the semantics more towards the auxiliary data, and is also at the risk of damaging the embedded semantics' accuracy, as KMR illustrates. Fig. 7(a) and Fig. 7(b) provides the impact of $P$ and $\beta$. Both of these factors are non-significant. Since $P$ is related to computation complexity, choosing a smaller $P$, like $P = 2$ can balance the efficiency and performance. For the momentum coefficient $\beta$, using the default setting $\beta = 0.9$ yields well-balanced performance on different models and metrics.

## 4 Related Work

**Large Vision Language Models.** Transformer-based LVLMs learn visual-semantic representations from large-scale image-text data, enabling tasks like image captioning [34, 14, 7, 37], visual QA [27, 32], and cross-modal reasoning [39, 28, 38]. Open-source models such as BLIP-2 [21], Flamingo [2], and LLaVA [24] show strong benchmark performance. Commercial models like GPT-4o, Claude-3.5 [4], and Gemini-2.0 [36] offer advanced reasoning and real-world adaptability, with their successors, GPT-o3 [30], Claude 3.7-Sonnet [3], and Gemini-2.5-Pro, able to reason in the text modality and vision modality.

**LVLM transfer-based attack.** Black-box attacks include query-based [10, 16] and transfer-based [11, 25] methods; this work focuses on the latter. AttackVLM [41] introduced transfer-based targeted attacks on LVLMs using CLIP [33] and BLIP [21] as surrogates, showing that image-to-image feature matching outperforms cross-modal optimization, a strategy adopted by later works [6, 13, 8, 22]. CWA [6] and SSA-CWA [8] applied this principle to commercial models like Bard [36], with CWA enhancing transferability via sharpness-aware minimization [12, 5], and SSA-CWA introducing spectrum-guided augmentation via SSA [26]. AnyAttack [40] utilizes image-image matching through large-scale pertaining and a subsequent fine-tuning. AdvDiffVLM [13] embeds feature matching into diffusion guidance, introduces Adaptive Ensemble Gradient Estimation (AEGE) for smoother ensemble scores. However, M-Attack outperforms these methods by a large margin through a simple local-level matching framework and an ensemble with diverse path sizes.

## 5 Conclusion

We find that M-Attack suffers from unstable gradients and identify the root causes as high variance and overlooked asymmetric matching. To this end, we introduce a principled framework that includes Multi-Crop Alignment (MCA) for variance reduction, Auxiliary Target Alignment (ATA) for semantic consistency, and Patch Momentum (PM) for replay-based stabilization. Combined with a refined surrogate model ensemble (PE$^+$), these components form M-Attack-V2, which achieves state-of-the-art results across multiple black-box LVLMs. We hope this study provides practical insights and encourages further research into stable and transferable adversarial optimization under realistic black-box constraints.

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
