# OpenReview forum: "M-Attack-V2: Pushing the Frontier of Black-Box LVLM Attacks via Fine-Grained Detail Targeting"
_NeurIPS.cc/2025/Conference — Submitted to NeurIPS 2025_

### Official Review · Reviewer_R2xh · 2025-06-22

**Clarity:** 4
**Significance:** 3
**Originality:** 3
**Rating:** 4
**Confidence:** 3

**Summary:**

The paper introduces M-Attack-V2, a gradient denoising framework. It has key components: Multi-Crop Alignment (MCA) averages gradients from diverse local transformations to cut variance and enhance convergence. Aux Target Alignment (ATA) aligns adversarial examples with both the main target and semantically related auxiliary samples, creating a smooth path in the embedding space. Momentum replay stabilizes and amplifies perturbations by maintaining gradient direction across local perturbation manifolds.

**Questions:**

1. Have you evaluated M-Attack-V2 on OOD datasets to verify its performance, given the framework's reliance on surrogate model consistency and auxiliary sample similarity that may fail in OOD scenarios?
2. The works lacks specific comparisons of computational complexity between M-Attack-V2 and other black-box LVLMs attack methods (e.g., AttackVLM, M-Attack).
3. In Line 83, the word "Supursingly" is misspelled.
4. In line 103, there is an extra letter "d" at the beginning of the sentence, which should be removed.

The motivation of the article is quite reasonable, and it is hoped that the authors can further explain how to address potential issues in specific deployment scenarios.

**Ethical Concerns:**

["NO or VERY MINOR ethics concerns only"]

**Final Justification:**

The motivation of this work is interesting. The authors have satisfactorily addressed my concern regarding computational overhead, I appreciate this clarification. But their performance may degrade in OOD scenarios, which slightly limits the method’s generalizability. Given the above, I will maintain my original score.

**Limitations:**

Yes.

**Paper Formatting Concerns:**

No.

**Quality:**

3

**Strengths And Weaknesses:**

**Strengths**: This work proposed to address the gradient instability in black-box adversarial attacks on Large Vision - Language Models (LVLMs). It innovatively uses modular components: one part reduces gradient variance and enhances convergence by averaging gradients from diverse local transformations, another aligns adversarial examples with both the main target and semantically related auxiliary samples to create a smooth path in the embedding space.

**Weakness**

1. **Out-of-Distribution Data**: M-Attack-V2 depends on surrogate model consistency and semantic similarity of auxiliary samples. OOD data may disrupt these, causing unstable gradient estimation and reduced attack effectiveness. This paper lacks OOD validation.
2. **Higher Computational Cost**: Generating auxiliary images for ATA and processing multiple crops in MCA increase computational overhead, making it less suitable for resource-constrained environments.

---

> ### Author Rebuttal · Authors · 2025-07-30
>
> We sincerely thank the reviewer for the constructive and valuable feedback, which will definitely help us improve the quality of our paper. We will accommodate all suggestions into the revised manuscript. Below, we provide detailed responses to each of the reviewer's comments.
>
> >**W1 & Q1**: Out-of-Distribution Data: M-Attack-V2 depends on surrogate model consistency and semantic similarity of auxiliary samples. OOD data may disrupt these, causing unstable gradient estimation and reduced attack effectiveness. This paper lacks OOD validation. Q1: Have you evaluated M-Attack-V2 on OOD datasets to verify its performance, given the framework's reliance on surrogate model consistency and auxiliary sample similarity that may fail in OOD scenarios?
>
> Following the suggestion, we have added experiments on finding auxiliary images in one OOD dataset, PatternNet, which is composed of remote sensing images with an overview look only. The results are presented in the table below, which confirms the reviewer's insights. Specifically, the ASR on Claude decreases by 21%. This ablation study highlights the importance of finding in-distribution data. However, our ATA framework is flexible and does not limit the method to find auxiliary data, so one can also refer to diffusion models with control methods to generate in-distribution images with the same semantics but different details.
>
> ---
> *Experiments on ATA OOD Data*:
>
> | Model|COCO KMR (a/b/c) | COCO ASR | Pattern KMR (a/b/c) | Pattern ASR |
> |-----------|:---------:|:--------:|:-------------:|:-----------:|
> | GPT‑4o              | 0.91 / 0.78 / 0.40 | 0.99 | 0.86 / 0.66 / 0.25 | 0.97 |
> | Claude 3.7‑thinking | 0.56 / 0.32 / 0.11 | 0.67 | 0.33 / 0.20 / 0.06 | 0.46 |
> | Gemini 2.5‑Pro      | 0.87 / 0.72 / 0.22 | 0.97 | 0.86 / 0.62 / 0.14 | 0.90 |
>
> *COCO denotes COCO train set, which is the original setting of the paper, Pattern denotes PatterNet, an OOD dataset which contains remote sensing images (only overhead view of the objects)*.
>
> ---
>
>
> >**W2 & Q2**: Higher Computational Cost: Generating auxiliary images for ATA and processing multiple crops in MCA increase computational overhead, making it less suitable for resource-constrained environments. Q2: The works lacks specific comparisons of computational complexity between M-Attack-V2 and other black-box LVLMs attack methods (e.g., AttackVLM, M-Attack).
>
> We acknowledge your concern about computational cost. We have provided a detailed theoretical analysis along with a comparison of empirical run time per image with other baseline methods, please see our response to *Q2* for Reviewer URye (the second reviewer). For quick reference, the overhead is flexible by configuring different values of $K$ and $P$. $K=2, P=2$ already provides sufficient improvement (+20%, +17%, +3% and +13% on Claude 3.7's $\text{KMR}_a$, $\text{KMR}_b$, $\text{KMR}_c$ and ASR) while only increasing slightly on per image time by 9.4% compared to $\texttt{M-Attack-V1}$, from 22.04 secons to 24.03 seconds per image.
>
> >**Q3 & Q4**: In Line 83, the word "Supursingly" is misspelled.
> In line 103, there is an extra letter "d" at the beginning of the sentence, which should be removed.
>
> Thanks for pointing the typos out. We have corrected them in the revised manuscript.
>
>
> >The motivation of the article is quite reasonable, and it is hoped that the authors can further explain how to address potential issues in specific deployment scenarios.
>
> We are encouraged by the kind comments. We have revised our paper carefully following the mentioned suggestions. In a real deployment scenario with even more constrained resources, we can further combine our $\texttt{M-Attack-V2}$ with other accleration techniques, such as Automatic Mixed Precision (AMP), FlashAttention, and static graph optimization, like `torch.compile`, to further reduce the overhead and accelerate the optimization process. In even more extreme cases, quantized models can be utilized as surrogate models to further reduce the overhead.

---

> > ### Comment · Reviewer_R2xh · 2025-08-02
> >
> > I thank the authors for their response, which has addressed most of my concerns. However, the additional OOD result suggest that performance may degrade in such settings, which slightly limits the method’s generalizability—an important consideration for real-world attack scenarios. Therefore, I will maintain my score.

---

> > > ### Author Response · Authors · 2025-08-04
> > > **Thanks for the further comments**
> > >
> > > We appreciate the reviewer for the further comments. We understand the reviewer's point that the additional OOD results suggest performance may degrade in OOD setting. In fact, this aligns with both our expectation and the reviewer's reasoning, as we used **remote sensing** data (the Pattern dataset) as an auxiliary supervision signal for the rebuttal, while the target dataset is COCO, which contains natural images. This verifies the reviewer described in the question: "OOD data may disrupt these, causing unstable gradient estimation and reduced attack effectiveness."
> > >
> > > To further validate this perspective, here we provide an in-distribution result as follows:
> > >
> > > | Model    | COCO  KMR (a/b/c) | COCO  ASR | COCO$_\text{auxiliary}$ KMR (a/b/c) | COCO$_\text{auxiliary}$  ASR |
> > > |-----------|:---------:|:--------:|:-------------:|:-----------:|
> > > | GPT‑4o              | 0.86 / 0.73 / 0.38 | 0.95 | 0.91 / 0.78 / 0.40 |0.99 |
> > > | Claude 3.7‑thinking | 0.55 / 0.22 / 0.10 | 0.62 | 0.56 / 0.32 / 0.11 | 0.67 |
> > > | Gemini 2.5‑Pro      | 0.85 / 0.68 / 0.21 | 0.93 | 0.87 / 0.72 / 0.22 | 0.97 |
> > >
> > > The final outcomes are consistent with expectations:
> > > 1. If *the auxiliary data and target data are from different domains*, for instance, one is **natural images** and the other is **remote sensing images** (as used in our rebuttal), the auxiliary data may provide inconsistent optimization directions, leading to decreased attack accuracy. Moreover, in real-world attack scenarios, we would likely not use remote sensing images to assist attacks on natural images, instead, we would follow the approach shown below.
> > > 2. If *the auxiliary data and target data are from the same domain* of **both natural images** in our paper, the auxiliary data can facilitate the optimization process and improve attack accuracy.
> > >
> > > If the reviewer has any further comments, we welcome continued discussion and are happy to provide additional clarifications.

---

### Official Review · Reviewer_DYji · 2025-06-24

**Clarity:** 3
**Significance:** 3
**Originality:** 4
**Rating:** 4
**Confidence:** 4

**Summary:**

This paper introduces M-Attack-V2, an improved transfer-based black-box adversarial attack on LVLMs. Building on M-Attack, the authors introduce three core components to stabilize gradient estimation:

- MCA: Averages gradients over multiple local crops to reduce variance.

- ATA: Regularizes with gradients from semantically similar auxiliary images

- PM: Reinterprets momentum as replay over past crop-level gradients, maintaining directional consistency despite stochastic local views.

Empirical evaluation on four commercial LVLMs (GPT-4o, Claude-3.7, Gemini-2.5-Pro, LLaVA-1.5) are promising over 100 testing images.

**Questions:**

1. Can we have a more comprehensive evaluation with more images covering more real-world domains (e.g., medical, satellite)?
2. Can we have a theoretical or empirical analysis of computational overhead of the proposed method?
3. Are the perturbations imperceptible?  Can we have an user studies to confirm the imperceptibility?
4. Since the M-Attack-V2 aims to introduce perturbation to the clean image, can input preprocessing serve as a defence? I suggest the authors to try some common method such as JPEG compression or DiffPure [1]


[1] Nie, Weili, et al. "Diffusion models for adversarial purification." arXiv preprint arXiv:2205.07460 (2022).

**Ethical Concerns:**

["NO or VERY MINOR ethics concerns only"]

**Final Justification:**

Overall, I find this paper present a good progress towards LVLM adversarial black-box attacks.

The rebuttal is good. My only concern is about its application in the real-world scenario as discussed.

Therefore, I keep my original ratting as borderline accept.

**Quality:**

4

**Strengths And Weaknesses:**

### **Strengths**

- The research problem on black-box LVLMs is interesting and critical.
- The proposed method is well motivated by analysing the weakness of M-Attack on the high-variance gradient estimates. The three proposed modules are reasonable and intuitive.
- The M-Attack-2 demonstrates strong empirical gains while it is efficient with early-stopping reducing query counts by 20–30% compared to baselines.

### **Weaknesses**

- Limited Dataset Scope: Evaluation on only 100 images (NIPS 2017 + COCO) may not reflect performance on diverse, real-world domains (e.g., medical, satellite).
- MCA and ATA can introduce computational overhead.
- Little analysis of how auxiliary selection quality or size impacts ATA’s effectiveness. If this data is not carefully prepared, how this noisy auxiliaries might degrade the attack's performance.
- Lack of qualitative comparisons or visualisation to clean image. I am curious if the optimised perturbations are still invisible compared to other attacks or even the clean image? I believe an user studies can help confirm the imperceptibility.
-  Defence and Perceptual Analysis Missing. No experiments against simple input preprocessing defences (e.g., JPEG compression)

---

> ### Author Rebuttal · Authors · 2025-07-30
>
> We sincerely appreciate the reviewer for the constructive and valuable feedback, which will definitely help us improve the quality of our paper. We will accommodate all suggestions into the revised manuscript. Below, we provide detailed responses to each of the reviewer's comments.
>
> >**W1 & Q1**: Limited Dataset Scope: Evaluation on only 100 images (NIPS 2017 + COCO) may not reflect performance on diverse, real-world domains (e.g., medical, satellite). Q1: Can we have a more comprehensive evaluation?
>
> Thank you for your suggestion. In our original submission, we included experiments on 1000 images of NIPS 2017 competition, presented in Table 10 *in the appendix*. Here we have also incorporated images from additional domains into our work. Specifically, we utilized *ChestMNIST*, a chest X-ray dataset from the *MedMNIST* [1], and *PatternNet* [2], a collection of overview remote sensing images. We augmented the original *NIPS2017 adversarial competition dataset* with these datasets while keeping the target set unchanged, formulating a cross-domain image attack, since current commercial LLMs struggle to describe these data (i.e., for different chest X-rays, they primarily respond with 'it is a chest X-ray' without specific diseases). The results are presented in the following table. Our method consistently outperforms M-Attack-V1 and other baselines. The gap on Claude 3.7 is even larger than that of the original NIPS 2017 dataset, demonstrating the superiority of our M-Attack-V2.
>
> *PatternNet*:
> |Method|GPT‑4o(a/b/c, ASR)|Claude 3.7 (a/b/c, ASR)|Gemini 2.5 (a/b/c, ASR)|
> |-|:-:|:-:|:-:|
> |AttackVLM|0.06/0.01/0.00 (0.02)| 0.06/0.02/0.00 (0.00)|0.09/0.04/0.00 (0.03)|
> |SSA‑CWA|0.05/0.02/0.00 (0.13)| 0.04/0.03/0.00 (0.07)|0.08/0.02/0.01 (0.15)|
> |AnyAttack|0.06/0.03/0.00 (0.05)| 0.03/0.01/0.00 (0.05)|0.06/0.02/0.00 (0.05)|
> |M‑Attack-V1|0.79/0.66/0.21 (0.93)|0.33/0.17/0.04 (0.48)|0.86/0.71/0.23 (0.91)|
> |M‑Attack-V2|0.83/0.71/0.24 (0.93)|0.58/0.40/0.09 (0.73)|0.88/0.68/0.22 (0.97)|
>
> *ChestMNIST*:
> |Method|GPT‑4o(a/b/c, ASR)|Claude 3.7 (a/b/c, ASR)|Gemini 2.5 (a/b/c, ASR)|
> |-|:-:|:-:|:-:|
> |AttackVLM|0.06/0.01/0.00 (0.03)|0.05/0.02/0.00 (0.02)|0.08/0.03/0.00 (0.02)|
> |SSA‑CWA|0.06/0.03/0.00 (0.15)|0.04/0.03/0.00 (0.07)|0.08/0.02/0.01 (0.14)|
> |AnyAttack|0.06/0.02/0.00 (0.05)|0.03/0.01/0.00 (0.04)|0.07/0.02 / 0.00 (0.05)|
> |M‑Attack-V1|0.89/0.70/0.22 (0.92)|0.31/0.18/0.07 (0.31)|0.85/0.67/0.23 (0.96)|
> |M‑Attack-V2|0.90/0.74/0.27 (0.97)|0.70/0.51/0.21 (0.83)|0.89/0.76/0.33 (0.95)|
>
> >**W2 & Q2**: MCA and ATA can introduce computational overhead. Q2: Can we have a theoretical or empirical analysis?
>
> Your insights are on point. Considering the performance gain they bring, these overheads are reasonable. Please also see our response to *Q2* for Reviewer URye (the second reviewer) for detailed theoretical and empirical analysis. For quick reference, due to GPUs' modern architecture, this parallelizable overhead is not as significant as it looks. For example, with K=2, P=2, per image time increase by 9.4% compared to M-Attack-V1, from 22.04 seconds to 24.13 seconds, whlie achieveing sufficient improvement (+20%,+17%,+3%, and +13% on Claude 3.7's $\text{KMR}\_a$, $\text{KMR}\_b$, $\text{KMR}\_c$ and ASR). Especially compared to SSA-CWA, which also utilizes a multi-sampling process, but takes $545.80\pm 4.21$ seconds and is outperformed by our method by a large margin, our improvement comes at a cost, but in an efficient way.
>
> >**W3**: Little analysis of how auxiliary selection quality or size impacts ATA’s effectiveness. If this data is not carefully prepared, how this noisy auxiliaries might degrade the attack's performance.
>
> Following the suggestion, we have conducted an ablation study by corrupting the ATA's image by adding Gaussian noise under different scaling factors. The results are presented in the table below. These corrupted auxiliary images cause an adverse effect. Thus, the introduction of the ATA applies additional requirements on the quality of the retrieved images. However, the ATA framework remains flexible: when only low‑quality or heavily noisy images are available, diffusion models can be leveraged to synthesize clean new images that preserve the same semantics. Additionally, we conduct similar experiments to test the performance with out-of-distribution data. Please see our response to *W1 & Q1* for Reviewer R2xh (the last reviewer) if interested.
>
> *Experiments on ATA Noisy Data*:
>
> | Model|KMR (0)|ASR (0)|KMR (0.5)|ASR (0.5)|KMR (1)|ASR (1)|
> |-|:-:|:-:|:--:|:-:|:-:|:-:|
> |GPT‑4o|0.91/0.78/0.40|0.99|0.84/0.66/0.21|0.98|0.90/0.72/0.28|0.98|
> |Claude 3.7‑thinking|0.56/0.32/0.11|0.67|0.59/0.35/0.10|0.69|0.47/0.27/0.05|0.61|
> |Gemini 2.5‑Pro|0.87/0.72/0.22|0.97|0.88/0.64/0.19|0.94|0.88/0.70/0.20|0.94|
>
> *Values inside brackets denote the scale of random noise.*
>
> >**W4 & Q3**: Lack of qualitative comparisons or visualisation to clean image. I am curious if the optimised perturbations are still invisible compared to other attacks or even the clean image? I believe a user study can help confirm the imperceptibility. Q3: Can we have a user study?
>
> Thank you for the constructive suggestion. We conducted two user studies to evaluate the imperceptibility of our method in comparison to the original images and to other baseline attack methods.
> - In the first study, users were shown 20 images (10 perturbed by M-Attack-V2, 10 unmodified) and asked to label each as perturbed or not, simulating a real-world blind setting
> - The second study showed 40 images (10 each from AnyAttack, SSA-CWA, M-Attack-V1, and M-Attack-V2), asking users to select the 20 they believed were corrupted, comparing the perceptual stealth of different methods (excluding VLM due to low performance)
>
> All adversarial images were generated under $\epsilon = 16$. The results, averaged over five distinct user groups, are summarized below. In general, only 42% of the adversarial images of M-Attack-V2 are identified, leaving 58% of the images passing the human check. Among all the tested methods, AnyAttack is the most identifiable. The SSA-CWA follows the second. M-Attack-V1 and M-Attack-V2 share around 30%. The first user study shows the potential threat of M-Attack-V2 under a real-world scenario, even with human supervision.
> | Outcome|Mean ± Std. (%)|
> |-|:-:|
> |Identified adversarial images|42 ± 1.7|
> |Identified original images|98 ± 1.6|
>
> |Method|Identified as Perturbated Images (%)|
> |-|:--:|
> |AnyAttack|84 ± 4.47|
> |SSA-CWA|54 ± 8.49|
> |M-Attack-V1|30 ± 10.1|
> |M-Attack-V2|32 ± 8.0|
>
> >**W5 & Q4**: Defence and Perceptual Analysis Missing. No experiments against simple input preprocessing defences (e.g., JPEG compression). Q4: Since the M-Attack-V2 aims to introduce perturbation to the clean image, can input preprocessing serve as a defence?
>
> Following the suggestion, we added experiments under JPEG (quality=75) and DiffPure defenses. For DiffPure, we followed the original setup but tested with $t = 25$ and $t = 75$ to reflect mild and strong purification (original $t=150$). However, even at $t=75$, we observed structural loss and artifacts (e.g., blurry or broken edges).
>
> Our results demonstrate that M-Attack-V2 remains robust under JPEG compression, while other methods, such as AttackVLM, SSA-CWA, and AnyAttack experience a substantial decline in performance, indicating that our method generates more resilient perturbations against low-level pixel transformations. When evaluated under DiffPure with $t=25$, all methods' performance drops, showing the capability of diffusion-based defense. However, M-Attack-V2 maintains an advantage, achieving significantly higher performance. Under DiffPure with $t=75$, where images become nearly regenerated, all methods exhibit low performance while  M-Attack-V2 still consistently outperforms other baselines.
>
> *Experiments on JPEG Defense*:
> |Method| GPT‑4o (a/b/c, ASR)|Claude 3.7 (a/b/c, ASR)|Gemini 2.5 (a/b/c, ASR)|
> |-|:-:|:-:|:-:|
> |AttackVLM|0.06/0.02/0.00 (0.03)|0.07/0.02/ 0.00 (0.02)|0.08/0.04/0.00 (0.04)|
> |SSA‑CWA|0.08/0.04/0.01 (0.10)|0.07/0.02/0.00 (0.05)|0.09/0.06/0.01 (0.09)|
> |AnyAttack|0.06/0.03/0.00 (0.05)|0.04/0.01/0.00 (0.03)|0.08/0.03/0.00 (0.05)|
> |M‑Attack-V1|0.76/0.54/0.16 (0.91)| 0.28/0.17/0.03 (0.34)|0.75/0.51/0.11 (0.76)|
> |M‑Attack-V2|0.89/0.69/0.20 (0.97)| 0.55/0.36/0.09 (0.68)|0.75/0.56/0.18 (0.82)|
>
> *Experiments on  Diffpure t = 25*:
> |Method|GPT‑4o (a/b/c, ASR)|Claude 3.7 (a/b/c, ASR)|Gemini 2.5 (a/b/c, ASR)|
> |-|:-:|:-:|:-:|
> |AttackVLM|0.05/0.02/0.00 (0.01)|0.05/0.02/0.00 (0.01)|0.08/0.03/0.00 (0.01)|
> |SSA‑CWA|0.07/0.03/0.00 (0.02)|0.04/0.02/0.00 (0.03)|0.07/0.01/0.00 (0.05)|
> |AnyAttack|0.07/0.03/0.00 (0.04)|0.02/0.02/0.00 (0.04)|0.09/0.04/0.00 (0.07)|
> |M‑Attack-V1|0.42/0.20/0.03 (0.43)|0.10/0.05/0.01 (0.10)|0.39/0.22/0.01 (0.32)|
> |M‑Attack-V2|0.73/0.47/0.15 (0.72)|0.19/0.11/0.04 (0.20)|0.61/0.42/0.06 (0.56)|
>
> *Experiments on Diffpure t = 75*:
> |Method| GPT‑4o (a/b/c, ASR)|Claude 3.7 (a/b/c, ASR)|Gemini 2.5 (a/b/c, ASR)|
> |-|:-:|:-:|:-:|
> |AttackVLM|0.08/0.05/0.00 (0.02)|0.04/0.02/0.00 (0.00)|0.04/0.01/0.00 (0.01)|
> |SSA‑CWA|0.05/0.03/0.01 (0.06)|0.05/0.03/0.00 (0.03)|0.07/0.02/0.00 (0.05)|
> |AnyAttack|0.05/0.00/0.00 (0.06)|0.04/0.02/0.00 (0.03)|0.04/0.02/0.00 (0.07)|
> |M‑Attack-V1|0.10/0.02/0.00 (0.04)|0.03/0.02/0.00 (0.02)|0.05/0.05/0.00 (0.05)|
> |M‑Attack-V2|0.13/0.06/0.01 (0.07)|0.07/0.02/0.00 (0.06)|0.12/0.06/0.01 (0.08)|
>
> -------------
> **References:**
>
> [1] Jiancheng Yang, Rui Shi, Donglai Wei, Zequan Liu, Lin Zhao, Bilian Ke, Hanspeter Pfister, Bingbing Ni. Yang, Jiancheng, et al. "MedMNIST v2-A large-scale lightweight benchmark for 2D and 3D biomedical image classification." Scientific Data, 2023.
>
> [2] Zhou W, Newsam S, Li C, et al. PatternNet: A benchmark dataset for performance evaluation of remote sensing image retrieval[J]. ISPRS journal of photogrammetry and remote sensing, 2018, 145: 197-209.

---

> > ### Comment · Reviewer_DYji · 2025-08-06
> > **Response to the rebuttal**
> >
> > I appreciate authors' rebuttal.
> >
> > Generally, with the additional results, the proposed attack is demonstrated to be significantly effective compared to the previous attacks.
> >
> > My only concern is about its application in the real-world scenario. The corrupted images still can be detected to some significant extent (e.g., 42% in the user study setup 1)
> >
> > Overall, I would keep my original rating.

---

> > > ### Author Response · Authors · 2025-08-06
> > >
> > > We appreciate the reviewer for your kind response and for recognizing the effectiveness of our attack approach.
> > >
> > > We fully understand the concern regarding real-world scenario applicability, particularly the perceptibility of perturbations to the human eye, a challenge faced by nearly all current adversarial attack methods. One practical solution is to reduce the perturbation budget ($\epsilon$), thereby minimizing visual artifacts.
> > >
> > > As shown in the table below, when $\epsilon$ is reduced from 16 to 8 and 4, our method still significantly outperforms prior competitive approaches in attack success rate, demonstrating its superior robustness under stricter, more realistic settings. This suggests that our approach is better suited for real-world deployment scenarios where imperceptibility is critical.
> > >
> > >
> > > | Method | GPT-4o (a/b/c,ASR) $\epsilon=4$ | Claude 3.7 (a/b/c,ASR) $\epsilon=4$ | Gemini 2.5 (a/b/c,ASR) $\epsilon=4$| GPT-4o (a/b/c,ASR) $\epsilon=8$ | Claude 3.7 (a/b/c,ASR) $\epsilon=8$ | Gemini 2.5 (a/b/c,ASR) $\epsilon=8$| GPT-4o (a/b/c,ASR) $\epsilon=16$ | Claude 3.7 (a/b/c,ASR) $\epsilon=16$| Gemini 2.5 (a/b/c,ASR) $\epsilon=16$ |
> > > |:-|:-:|:-:|:-:|:-:|:-:|:-:|:-:|:-:|:-:|
> > > | AttackVLM [1] | 0.08/0.04/0.00 (0.02) | 0.04/0.01/0.00 (0.00) | 0.10/0.04/0.00 (0.01) | 0.08/0.02/0.00 (0.01) | 0.04/0.02/0.00 (0.01) | 0.07/0.01/0.00 (0.01) | 0.08/0.02/0.00 (0.02) | 0.01/0.00/0.00 (0.01) | 0.03/0.01/0.00 (0.00) |
> > > | SSA-CWA [2] | 0.05/0.03/0.00 (0.03) | 0.04/0.01/0.00 (0.02) | 0.04/0.01/0.00 (0.04) | 0.06/0.02/0.00 (0.04) | 0.04/0.02/0.00 (0.02) | 0.02/0.00/0.00 (0.05) | 0.11/0.06/0.00 (0.09) | 0.06/0.04 / 0.01 (0.12) | 0.05/0.03/0.01 (0.08) |
> > > | AnyAttack [3] | 0.07/0.02/0.00 (0.05) | 0.05/0.05/0.02 (0.06) | 0.05/0.02/0.00 (0.10) | 0.17/0.06/0.00 (0.13) | 0.07/0.07/0.02 (0.05) | 0.12/0.04/0.00 (0.13) | 0.44/0.20/0.04 (0.42) | 0.19/0.08/0.01 (0.22) | 0.35/0.06/0.01 (0.34) |
> > > | **$\texttt{M-Attack-V2}$** | **0.59/0.34/0.10 (0.58)** | **0.06/0.02/0.00 (0.02)** | **0.48/0.33/0.07 (0.38)** | **0.87/0.69/0.20 (0.93)** | **0.23/0.14/0.02 (0.22)** | **0.72/0.49/0.21 (0.77)** | **0.91/0.78/0.40 (0.99)** | **0.56/0.32/0.11 (0.67)** | **0.87/0.72/0.22 (0.97)** |
> > >
> > >
> > >
> > > Moreover, we provide a user study breakdown, which shows that the proportion of adversarial images correctly identified by users drops considerably as $\epsilon$ decreases. We also report a confusion rate between adversarial and real images, which improves accordingly, further supporting the potential of our method in practical settings.
> > >
> > >
> > > | Proportion                           | $\epsilon=16$, Mean ± Std  | $\epsilon=8$, Mean ± Std |
> > > |-----------------------------------|:------------------:|:------------------:|
> > > | Adversarial images identified by users $\downarrow$  | 42 ± 1.7    |   27.4 ± 1.6   |
> > > | Confusion rate between adversarial and real images $\downarrow$  | 98 ± 1.6    |  93.1 ± 2.3   |
> > >
> > >
> > > We will clarify this further in our revised paper. Thank you again for your valuable feedback.
> > >
> > > ----------
> > > References:
> > >
> > > [1] Zhao Y, Pang T, Du C, et al. On evaluating adversarial robustness of large vision-language models[J]. Advances in Neural Information Processing Systems, 2023, 36: 54111-54138.
> > >
> > > [2] Dong Y, Chen H, Chen J, et al. How robust is google's bard to adversarial image attacks?[J]. arXiv preprint arXiv:2309.11751, 2023.
> > >
> > > [3] Zhang J, Ye J, Ma X, et al. AnyAttack: Towards Large-scale Self-supervised Adversarial Attacks on Vision-language Models[C] Proceedings of the Computer Vision and Pattern Recognition Conference. 2025: 19900-19909.

---

### Official Review · Reviewer_URye · 2025-06-30

**Clarity:** 3
**Significance:** 3
**Originality:** 3
**Rating:** 5
**Confidence:** 3

**Summary:**

This work identifies and addresses the issue of high-variance gradient estimates in black-box adversarial attacks on LVLMs that arise from using a local crop matching strategy in previous *M-Attack* method. The authors propose **M-Attack-V2**, a framework that incorporates *Multi-Crop Alignment (MCA)* to reduce variance, *Auxiliary Target Alignment (ATA)* to ensure semantic consistency, *Patch Momentum (PM)* to stabilize gradient optimization, and a refined surrogate model ensemble *Patch Ensemble+ (PE+)*.

**Questions:**

* The ablation study in *Tab. 4* indicates that removing **Patch Momentum (PM)** results in a far greater performance drop than removing MCA. Given that the paper's motivation heavily emphasizes the orthogonal gradient problem, which MCA is introduced to address, could you clarify why PM has a much more dominant effect on the overall performance? Does this suggest that gradient orthogonality might not be the primary factor limiting performance, as you initially framed?
* Have you compared the computational cost (e.g., time per image) between M-Attack and M-Attack-V2?

**Ethical Concerns:**

["NO or VERY MINOR ethics concerns only"]

**Final Justification:**

This work addresses an interesting and important problem in black-box adversarial attacks. The authors' rebuttal has successfully resolved my initial concerns, and I will maintain my score.

**Limitations:**

Yes, the limitations is in the appendix.

**Quality:**

3

**Strengths And Weaknesses:**

### **Strengths**
* The technical approach is meaningful and the motivation is strong. The authors demonstrate that the gradients computed from randomly sampled local crops are nearly orthogonal, leading to unstable optimization. Then they propose a framework named **M-Attack-V2** and have solved the problem well.
* The experiment is comprehensive. The authors test their method against multiple frontier commercial LVLMs and achieve significant performance improvement. They also have ablation studies on each component and key hyperparameters to show the effectiveness of their methods.
### **Weaknesses**
In this work, the analysis of the *Auxiliary Target Alignment (ATA)* remains at an intuitive level. The authors claim that ATA provides "low-variance, semantics-preserving updates" and "constructs a smooth semantic trajectory in the embedding space", but they offer no theoretical analysis to demonstrate how these properties are achieved.

---

> ### Author Rebuttal · Authors · 2025-07-30
>
> We sincerely appreciate the reviewer for the constructive and valuable feedback, which will definitely help us improve the quality of our paper. We will accommodate all suggestions into the revised manuscript. Below, we provide detailed responses to each of the reviewer's comments.
>
> >**W1**: In this work, the analysis of the Auxiliary Target Alignment (ATA) remains at an intuitive level. The authors claim that ATA provides "low-variance, semantics-preserving updates" and "constructs a smooth semantic trajectory in the embedding space", but they offer no theoretical analysis to demonstrate how these properties are achieved.
>
> Thanks for pointing out the significance of more theoretical analysis. We have provided a preliminary theoretical analysis of the Auxiliary Target Alignment (ATA) below. We will analyse further how it improves the optimization process and add the complete analysis to the revised manuscript.
>
> ***Theoretical Analysis of the ATA***
>
> We begin with three mild assumptions.
>
> ***Assumption 1 ($L$-continus):*** *Let $f$ denote the surrogate model which is $L$-continus. Then, for any two inputs $x,y$,*$$f(y) - f(x) \\le L \\lVert y-x \\rVert.\tag{1}$$
>
> ***Assumption 2 (bounded auxiliary data)***: *For auxiliary data $X\_{\\text{aux}}^{(p)}$, retrieved based on similarity to the target $X\_\\text{tar}$, we assume* $$\\mathbb{E}[\\lVert f(X^{(p)}\_{\\text{aux}})-f(X\_{\\text{tar}}) \\rVert ]\\le\\delta. \tag{2}$$
> This assumption stems from the construction:$$X\_{\\text{aux}}^{(p)} =\\arg \\mathrm{top}\_{X \\in \\mathcal{D}}^{p} \\frac{f(X)^\\top f(X\_{\\text{tar}})}{\\lVert f(X)\\rVert{\\lVert f(X\_{\\text{tar}}) \\rVert}} \\tag{3},$$where $\\arg\\mathrm{top}^p\_{X \\in \\mathcal{D}}$ denotes $p$-th highest matching item in the database $\\mathcal{D}$ by cosine similarity. Since embedding $f(X)$ is normalized by convention, $\\delta$ exists.
>
> ***Assumption 3 (bounded transformation):*** *Let $\\mathcal{T}\\sim D\_\\alpha$* denote a random transformation drawn from a distribution with bounded pixel-level distortion. Then$$\\mathbb{E}\_{T\\sim D\_{\\alpha}}[\\lVert \\mathcal{T}(X)-X \\rVert]\\le\\alpha.\tag{4}$$
>
> We define the *embedding drift* of a transformation $\\mathcal{T}$ appied to $X$ on model $f$ as $\\Delta\_{\\text{drift}}(\\mathcal{T};X) := \\mathbb{E}\_{\\mathcal{T}}[\\lVert f(\\mathcal{T}(X)) - f(X)\\rVert]$. We now state the main result.
>
> ***Theorem 1:*** *Let $\\mathcal{T}~\\sim D\_\\alpha$ be the transformation used in V1, $\\tilde{\\mathcal{T}} \\sim D\_{\\tilde{\\alpha}}$, with $\\tilde{\\alpha} \\ll \\alpha$ be transformation used in V2, we have*$$\\begin{aligned}& \\Delta\_{\\text{drift}}(\\mathcal{T};X)  \\le L\\alpha \\\\ & \\Delta\_{\\text{drift}}(\\tilde{\\mathcal{T}};X\_{\\text{aux}}^{(p)})  \\le L\\tilde{\\alpha} + \\delta\\end{aligned}\\tag{5}$$**Theorm 1** shows that the drift of $\\texttt{M-Attack-V1}$ is bounded by $L\\alpha$, while drift of $\\texttt{M-Attack-V2}$ is bounded additional by $\\delta$. The term $L\\alpha$ reflects the inherent asymmetry of the target branch transformation. Transformations applied in the pixel space require an additional multiplier $L$ to account for their effect in the embedding space, thereby guiding the source image $X\_{\\text{sou}}$. In contrast, the auxiliary data operate directly in embedding space with $\\delta$. In practice, estimation $\\delta$ is significantly easier than $\\alpha$, since both $L$ and $\\alpha$ are generally unknown and hard to quantify. Moreover, the auxiliary data obtained via Eq.(3) inherently ensures better semantic alignment with the target. Thus, V2 strikes a balance by operating under a lower pixel distortion $\\tilde{\\alpha} \\ll \\alpha$, preserving the shift budget for more meaningful exploration via $\\delta$, finding a sweet point between exploration and exploitation.
>
> *Proof:*
> For $\\Delta\_{\\text{shift}}$ of V1
> $$\\begin{aligned}\\Delta\_{\\text{drift}}(\\mathcal{T};X) & = \\mathbb{E}\_{\\mathcal{T}\\sim D\_{\\alpha}}[\\lVert f(\\mathcal{T}(X)) - f(X) \\rVert] \\quad \\text{(By Assumption 1)} \\\\ & \\le L \\cdot \\mathbb{E}\_{\\mathcal{T}\\sim D\_{\\alpha}}[\\mathcal{T}(X)-f(X)] \\quad \\text{(By Assumption 3)}\\\\ & \\le L\\alpha, \\end{aligned}\\tag{6}$$For $\\Delta\_{\\text{shift}}$ of V2
> $$\\begin{aligned} \\mathbb{E}\_{\\tilde{\\mathcal{T}}\\sim D\_{\\tilde{\\alpha}}}[\\lVert f(\\tilde{\\mathcal{T}}(X^{(p)}\_{\\text{aux}})-\\tilde{\\mathcal{T}}(X)) \\rVert] & \\le  \\mathbb{E}\_{\\tilde{\\mathcal{T}}\\sim D\_{\\tilde{\\alpha}}}[\\lVert f(\\tilde{\\mathcal{T}}(X\_{\\text{aux}}^{(p)}))-f(X\_{\\text{aux}}^{(p)} ) \\rVert  + \\lVert f(X\_{\\text{aux}}^{(p)})-f(X\_{\\text{tar}}) \\rVert ]
> \\\\ &=  \\mathbb{E}\_{\\tilde{\\mathcal{T}}\\sim D\_{\\tilde{\\alpha}}} [\\lVert f(\\tilde{\\mathcal{T}}(X\_{\\text{aux}}^{(p)}))-f(X\_{\\text{aux}}^{(p)} ) \\rVert] + \\mathbb{E}\_{\\tilde{\\mathcal{T}}}[\\lVert f(X\_{\\text{aux}}^{(p)}-f(X\_{\\text{tar}})) \\rVert]
>  \\\\ & \\le L \\cdot \\mathbb{E}\_{\\tilde{\\mathcal{T}}\\sim D\_{\\tilde{\\alpha}}}\\lVert \\tilde{\\mathcal{T}}(X^{(p)}\_{\\text{aux}}) - X^{(p)}\_{\\text{aux}} \\rVert + \\delta \\\\ & \\le L\\tilde{\\alpha} + \\delta. \\end{aligned} \\tag{7}$$
> Here, we utilized the triangular inequality, Assumption 1 (Lipschitz continuity), and Assumptions 2 and 3 to derive the bounds.
>
> ---
> >**Q1**: The ablation study in Tab. 4 indicates that removing Patch Momentum (PM) results in a far greater performance drop than removing MCA. Given that the paper's motivation heavily emphasizes the orthogonal gradient problem, which MCA is introduced to address, could you clarify why PM has a much more dominant effect on the overall performance? Does this suggest that gradient orthogonality might not be the primary factor limiting performance, as you initially framed?
>
> We apologize for the ambiguity of Tab. 4 in our paper. We clarify that PM consists of two parts. The second-to-last line in Tab. 4 removes the first-order PM for accumulation, and the last removes the second-order PM for scaling. First order PM accmulation gradient of different patches in diferent iterations, just like intra-iteration MCA, remove it only cause slight decrease, whereas remove the MCA and ATA ***reduce ASR more significantly*** than removing first-order PM (for instance, removing MCA and ATA cause -10% and -1% of $\\text{KMR}\_b$ on Gemini and Clude, while removing first-order PM only cause -1% and -0%).
>
> The second-order PM is used for the notorious scaling issue [1,2]. Previous works mainly utilize FGSM (like AttackVLM, SSA-CWA, and $\\text{M-Attack-V1}$) to address it without further explanation and ablation, i.e., apply a sign function $\\mathrm{sign}(\\nabla\_{X\_{\\text{sou}}}\\mathcal{L})$. If we keep the first-order PM and use FGSM for scaling, as shown in the appendix, ***the decrease drops significantly to a reasonable value*** (for example, from -49% to 29% to -2% to -8 on $\\text{KMR}\_b$ for Gemini and Claude). We intend to introduce this ablation ***to be the first to highlight a potential scaling issue that has been previously neglected***; however, without proper analysis and explanation in the paper, this creates unnecessary misunderstanding. Thanks for pointing this out. We have then refined our presentation to avoid causing such a misunderstanding.
>
> >**Q2**: Have you compared the computational cost (e.g., time per image) between M-Attack and M-Attack-V2?
>
> Yes, here we provide both a theoretical analysis of the computational cost for different methods, followed by a comparison of empirical run time.
>
> Let $d$ denote the hidden dimension, $d\_{\\mathrm{ff}} = 4d$ the feed‑forward expansion, and $N$ the sequence length. In one transformer layer, the FFN incurs $2Nd\\,d\_{\\mathrm{ff}} = 8Nd^{2}$ FLOPs, while multi‑head attention adds $4Nd^{2} + 4N^{2}d$, giving a forward cost $M := 12Nd^{2} + 4N^{2}d$. Because the backward pass is empirically twice the forward cost, a complete forward–backward iteration for the baseline VLM requires $3M$ FLOPs. Exhaustively counting FLOPs for every architecture in an $N$‑model ensemble is impractical, so we introduce
> $$\\rho\_{N} \\;\\triangleq\\; \\frac{\\text{per‑iteration FLOPs of the }N\\text{-model ensemble}}{\\text{per‑iteration FLOPs of one CLIP‑B/16}}, \\tag{8}$$ an empirically measured inflation factor ($\\rho\_{1}=1$). Under this convention
> - **AttackVLM** costs $3M$ flops.
> - **M‑Attack‑V1** costs $3\\rho\_{N}M$ FLOPs per iteration.
> - **SSA‑CWA** adds an inner sampling loop of $\\hat{K}$ steps for sharpeness aware minimization (SAM), lifting the complexity to $3\\rho\_{N}\\hat{K}M$.
> - **M‑Attack‑V2** evaluates $K$ local crops and forwards $P$ auxiliary examples per crop to reduce variance in $\\texttt{M-Attack-V1}$, resulting in $\\rho\_{N}K\\bigl(3M+P\\bigr)$ FLOPs.
>
> In practice, real-time cost per image differs from FLOPs due to GPU parallelism. On a single NVIDIA RTX 4090, SSA-CWA takes $545.80 \pm 4.21$s per image, while $\texttt{M-Attack-V1}$ takes $22.04 \pm 0.11$s. Our $\texttt{M-Attack-V2}$ is flexible; with $K=2, P=2$, it runs in $24.13 \pm 0.84$s per image, a 9.4% increase, yet yields sufficient performance gains already (+20%, +17%, +3%, +13% on Claude 3.7's $\text{KMR}_a$, $\text{KMR}_b$, $\text{KMR}_c$, and ASR; +8%, +3%, +1%, +6% on GPT-4o's counterparts). *We omit AttackVLM for its low performance and AnyAttack as it allocates most of the computational resources to pre-training of its autoencoder*.
>
> -------------
>
> **References:**
>
> [1] Yushun Zhang, Congliang Chen, Tian Ding, Ziniu Li, Ruoyu Sun, and Zhiquan Luo. Why transformers need adam: A hessian perspective. Advances in neural information processing systems, 37:131786–131823, 2024.
>
> [2] Frederik Kunstner, Jacques Chen, Jonathan Wilder Lavington, and Mark Schmidt. Noise is not the main factor behind the gap between sgd and adam on transformers, but sign descent might be. arXiv preprint arXiv:2304.13960, 2023.

---

> > ### Comment · Reviewer_URye · 2025-08-06
> >
> > Thank you for the response, you have clarified all my questions, and I will maintain my score.

---

### Official Review · Reviewer_zTNE · 2025-07-02

**Clarity:** 2
**Significance:** 3
**Originality:** 3
**Rating:** 3
**Confidence:** 1

**Summary:**

This paper focuses on the blakc-box attacks on existing MLLMs. It proposes a several solutions to mitigate the limitations in its V1 version. However, because there is a lack of problem formulation and introduction of their V1 work. The overall paper is extremely hard to follow.

**Questions:**

In general, after reading the paper, I felt more like I was reading an appendix or extension of the M-Attack paper, rather than an independent new paper. As a result, I can only assess this submission with the lowest confidence score. I strongly encourage the authors to include a more thorough introduction to M-Attack in their revised version, so that the paper can be better understood by readers who are not already familiar with that prior work.

**Ethical Concerns:**

["NO or VERY MINOR ethics concerns only"]

**Final Justification:**

I will maintain my current confidence and score. The paper lacks sufficient exposition. I understand it as an upgrade of M-Attack-V1, but the manuscript does not adequately motivate the problem or situate it within the relevant background, which makes the paper difficult to follow.

**Limitations:**

Yes

**Quality:**

2

**Strengths And Weaknesses:**

***Strength***:
1. The paper focuses on an important research problem.
2. The paper conductes comprehensive experiments, and achieves noticeable performance improvment.

***Weakness***:

*** A lack of Background and Problem Formulation for M-Attack makes the submission difficult to understand ***

The overall writing structure is somewhat unfriendly to readers who are not already familiar with M-Attack. I attempted to check the reference for M-Attack, but found that the citation is broken, making it difficult to trace the original work.

Here are some mild suggestions:  the authors should at least provide a concise technical summary of M-Attack (even in the appendix). More importantly, a rigorous research paper—even if it is a MARK II version—should include a dedicated subsection that clearly formulates the research problem and defines their threat model. While such context may become unnecessary if M-Attack is widely recognized as a standard baseline in this area, at present, the lack of background makes it hard for readers unfamiliar with M-Attack (like myself) to fully follow the paper’s motivation and methodology. For example, what are X_tar, X_sou and X_adv defined in the first paragraph of Method? What are image pairs and region pairs mentioned in the paper? Why include IoU in the motivation experiment?

---

> ### Author Rebuttal · Authors · 2025-07-30
>
> We sincerely thank the reviewer for the constructive and valuable comments, which will definitely help us improve the quality of our paper. We will accommodate all suggestions into the revised manuscript. Below, we provide detailed responses to each of the reviewer's questions.
>
> >**W1 & Q1**: *** A lack of Background and Problem Formulation for M-Attack makes the submission difficult to understand ***
> The overall writing structure is somewhat unfriendly to readers who are not already familiar with M-Attack. I attempted to check the reference for M-Attack, but found that the citation is broken, making it difficult to trace the original work.
> Here are some mild suggestions: the authors should at least provide a concise technical summary of M-Attack (even in the appendix). More importantly, a rigorous research paper—even if it is a MARK II version—should include a dedicated subsection that clearly formulates the research problem and defines their threat model. While such context may become unnecessary if M-Attack is widely recognized as a standard baseline in this area, at present, the lack of background makes it hard for readers unfamiliar with M-Attack (like myself) to fully follow the paper’s motivation and methodology.
> Q1: Felt more like reading an appendix or extension of the M-Attack paper. As a result, I can only assess this submission with the lowest confidence score. I strongly encourage the authors to include a more thorough introduction to M-Attack in their revised version, so that the paper can be better understood by readers who are not already familiar with that prior work.
>
> Thanks for your constructive feedback and suggestion, and we sincerely apologize for any inconvenience caused during your review. To address the concern, we have included a dedicated *preliminaries* section in the revised manuscript that provides a formal problem setup and clear definitions of all notations. For your convenience, we have also included the relevant content below. We hope this addition makes the paper more self-contained, allowing a broader audience to engage with our work without requiring too much prior knowledge.
>
> ### Preliminaries
>
> **Black-box Transfer Attack under Image-Image Matching**
>
> Let $f\_\\xi: \\mathrm{R}^{3\\times h\\times w} \\times \\mathcal{T} \\to \\mathcal{T}$ denote the target blackbox model that maps the image text pair to target text, with $h,w$ for height and width. Let $X\_\\text{sou}, X\_{\\text{tar}} \\in \\mathrm{R}^{3\\times h\\times w}$ denote the source and target image. The source image is clean at initial time, which we denote as $\\tilde{X}\_\\text{sou}$ , and $X\_\\text{sou} = \\tilde{X}\_\\text{sou} + \\delta$, with $\\delta=0$ initially. The ideal objective of a blackbox-transfer attack is to learn the perturbation $\\delta$ such that the perturbated source image matches the target image in the semantic feature space, i.e., $f\_\\xi(\\tilde{X}\_\\text{sou}+\\delta) = f\_\\xi(X\_\\text{tar})$. However, directly optimizing such a target is difficult due to the unknown black-box model $f\_\\xi$. Therefore, Zhao et. al. [1] proposed an alternative optimization target:
> $$
> \\begin{aligned}
> & \\arg \\max\_{X\_\\text{sou}} \\mathrm{CS}(f\_\\phi(X\_\\text{sou}), f\_\\phi(X\_\\text{tar}))  \\\\
> & \\ \\ \\mathrm{s.t.} \\ \\lVert \\delta \\rVert\_p \\le \\epsilon,
> \\end{aligned} \\tag{1}
> $$
> where $f\_\\phi$ denote the surrogate model and $\\mathrm{CS}(a,b)=a^\\top b/( \\lVert a\\rVert\_2\\lVert b\\rVert\_2)$ denotes cosine similarity.
>
> ***Local-level Matching From $\\texttt{M-Attack-V1}$***.
> $\\texttt{M-Attack-V1}$ further extends on Equ. (1) with the idea of 'local level matching via cropping'. Specifically, it defines local transformation $\\mathcal{T}\_s, \\mathcal{T}\_t$ for source and target images and corresponding local reagions in each iteration $\\{ \\hat{\\bf x}\_{1}^{s}, \\dots, \\hat{\\bf x}\_{n}^{s} \\} = \\mathcal{T}\_s(\\bf X\_{\\text{sou}})$; $\\{ \\hat{\\bf x}^{t}\_{1}, \\dots, \\hat{\\bf x}\_{n}^{t}\\}/\\{\\hat{\\bf x}^{t}\_g\\} = \\mathcal{T}\_{t}(\\bf X\_{\\text{tar}})$ which satisfy the two essential properties:
>
> $$
> \\begin{aligned}
>     &\\forall i,j, \\quad \\hat{\\bf x}\_i \\cap \\hat{\\bf x}\_j \\neq \\emptyset\\\\
>     & \\forall i,j, \\quad \\lvert \\hat{\\bf x}\_i \\cup \\hat{\\bf x}\_j \\rvert > \\lvert \\hat{\\bf x}\_i \\rvert \\ \\text{and} \\ \\lvert \\hat{\\bf x}\_i \\cup \\hat{\\bf x}\_j \\rvert > \\lvert \\hat{\\bf x}\_j \\rvert ,
> \\end{aligned} \\tag{2}
> $$
>
> where subscipt $i,j$ denote the local regions in $i,j$-th iteration. Then, it rewrites objective $\\mathrm{CS}(f\_\\phi(X\_\\text{sou}), f\_\\phi(X\_\\text{tar}))$ in Equ. (1) under the local-level matching framework as
> $$
> \\mathcal{M}\_{\\mathcal{T}\_s, \\mathcal{T}\_t} = \\mathbb{E}\_{ f\_{\\phi\_j} \\sim \\phi} \\left [ \\text{CS} \\left (f\_{\\phi\_j}(\\hat{\\bf x}^s\_i), f\_{\\phi\_j}(\\hat{\\bf x}^{t}\_i \\right) \\right ], \\tag{3}
> $$
> where $\\phi\_j$ is the surrogate model sampled from the surrogate ensemble $\\phi$. This modified framework only match a pair of local regions $(\\hat{\\bf x}\_s^i, \\hat{\\bf x}\_t^i)$ in each iteration $i$ instead of whole image $X\_\\text{sou}, X\_\\text{tar}$. Such local-level matching enhances the semantics of the perturbation, further aided by the ensemble models to combine semantic details from different models. Therefore, $\\texttt{M-Attack-V1}$ significantly outperforms other baseline methods. While effective, this approach suffers from inherent instability in gradient-based optimization. Specifically, since $\\nabla\_{\\bf X\_{\\text{sou}}}  \\mathcal{M}\_{\\mathcal{T}\_s, \\mathcal{T}\_t}$ varies significantly across $\\mathcal{T}\_s$ (same to $\\mathcal{T}\_t$), the stochastic gradients become nearly orthogonal, i.e., $\\langle \\nabla\_{ X\_{\\text{sou}}^i}  \\mathcal{M}\_{\\mathcal{T}\_s, \\mathcal{T}\_t}, \\nabla\_{X\_{\\text{sou}}^{i+1}}  \\mathcal{M}\_{\\mathcal{T}\_s, \\mathcal{T}\_t} \\rangle \\approx 0$, where $X\_{\\text{sou}}^i$ is the source image in $i$-th iteration during optimziation. Such low similarity leads to high variance and poor convergence during the optimization process. The following section provides a detailed illustration.
>
> [1] Zhao, T. Pang, C. Du, X. Yang, C. Li, N.-M. M. Cheung, and M. Lin. On evaluating adversarial robustness of large vision-language models. In International Conference on Advanced Neural Information Processing Systems, pages 54111–54138, 2023
>
> *We hope our newly added preliminaries can address your question. Once again, we apologize for the inconvenience during your review process.*
>
> In the following, we address each of the other mentioned mild suggestions one by one. We have also integrated these responses into the newly revised sections of our revision.
>
> >**W2**: For example, what are X_tar, X_sou and X_adv defined in the first paragraph of Method?
>
> $X\_\\text{sou}$ is the source image we attach a perturbation $\\delta$ to. $X\_\\text{sou}$ is initialized as clean image, for which we use $\\tilde{X}\_\\text{sou}$, and $X\_\\text{sou}=\\tilde{X}\_\\text{sou} + \\delta$, with $\\delta=0$ in the initial stage. $X\_\\text{tar}$ denotes target image. The target of black-box adversarial attack is to find suitable $\\delta$ to make perturbated $X\_\\text{sou}$ (also denoted as $X\_\\text{adv}$ in some other works) share the same semantic as the $X\_\\text{tar}$ on the target black-box model, within limited perturbation, i.e., $\\lVert \\delta \\rVert\_p \\le \\epsilon$, where $\\lVert \\cdot \\rVert$ dentos $\\ell\_p$ norm.
>
>
> >**W3**: What are image pairs and region pairs mentioned in the paper?
>
> The image pairs consist of the paired source and target images, i.e., $X\_\\text{sou}$ and $X\_\\text{tar}$. In each attack, a clean image $\\tilde{X}\_\\text{sou}$ is used to initialize the source image $X\_\\text{sou}$, and a target image $X\_\\text{tar}$ is selected with a distinct semantic that the source image aims to align with under the black-box model $f\_\\xi$. The region pairs are proposed in $\\texttt{M-Attack-V1}$ as part of the 'local-level matching via cropping', a key innovation in $\\texttt{M-Attack-V1}$. In iteration $i$, one local region (usually selected via cropping) $\\hat{\\bf x}^i\_s$ from $X\_\\text{sou}$ is matched with one region $\\hat{\\bf x}^i\_t$ from $X\_\\text{tar}$, to formulate a local-level matching framework.
>
> >**W4**: Why include IoU in the motivation experiment?
>
> Because the local regions $\\hat{\\bf x}^1\_t, \\hat{\\bf x}^2\_s,\\dots, \\hat{\\bf x}^i\_s$ vary in crop size, two large crops are statistically more likely to share a large intersection. Gradient similarity, however, is accumulated only over the overlapping pixels (non‑overlapping pixels are masked out). To remove this size-dependent bias, we plot the raw dot-product against Intersection-over-Union (IoU) instead of intersection size, obtaining a similarity measure that reflects genuine gradient alignment.

---

### Comment · Area_Chair_o9YL · 2025-08-05
**Please provide your feedback on the authors' rebuttal.**

Dear reviewers,

For those who have not responded yet, please take a look at the authors’ rebuttal and update your final scores.

Best wishes,

AC

---

### Note · Authors · 2025-08-12

Dear Area Chair and Reviewers,

We sincerely thank you for your insightful comments and constructive feedback. In our rebuttal, we have provided detailed responses to all suggestions, supported by new experiments and theoretical analysis. All new content will be incorporated into the revised manuscript.

We are encouraged that the reviewers recognized our work's key strengths. There is a consensus [zTNE, URye, DYji, R2xh] that our methods (MCA, ATA, PM) are well-motivated, yield substantial performance gains, and are validated by comprehensive experiments [zTNE, URye]. In response to specific feedback, we have provided more extensive evaluations for Reviewers DYji and R2xh and added a new theoretical study to further validate our Auxiliary Target Alignment (ATA) component [URye]. We trust that these additions, along with the follow-up response, will prove satisfactory.

Although we were unable to engage in further discussion with Reviewer zTNE, we have carefully considered the initial feedback regarding the lack of necessary background. To enhance the paper's accessibility, we have drafted a new "Preliminaries" section with a detailed and comprehensive background to make the work more self-contained for a broader audience. We hope that this addition, prepared in response to mild suggestions in the initial review, can successfully address this point.

Sincerely,

The Authors

---

### Decision · Program_Chairs · 2025-09-17

**Decision:**

Reject

**Comment:**

The reviews (3, 5, 4, 4) for this paper have been collected and discussed. There is a general consensus among the reviewers that the paper, in its current form, has some contributions to the field.

However, the primary concern raised by the reviewers is that “the lack of background and problem formulation for M-Attack makes the submission difficult to understand.” Requiring reviewers to have very specific prior knowledge in order to assess the work is problematic.

After carefully checking all the information and discussions among AC/SAC, the final recommendation is Rejection.